# Cardiomyocyte contractile impairment in heart failure results from reduced BAG3-mediated sarcomeric protein turnover

Thomas G. Martin[1], Valerie D. Myers[2], Praveen Dubey[2], Shubham Dubey[2], Edith Perez [1], Christine S. Moravec[3], Monte S. Willis[4], Arthur M. Feldman[2] & Jonathan A. Kirk [1]✉

The association between reduced myofilament force-generating capacity ($F_{max}$) and heart failure (HF) is clear, however the underlying molecular mechanisms are poorly understood. Here, we show impaired $F_{max}$ arises from reduced BAG3-mediated sarcomere turnover. Myofilament BAG3 expression decreases in human HF and positively correlates with $F_{max}$. We confirm this relationship using BAG3 haploinsufficient mice, which display reduced $F_{max}$ and increased myofilament ubiquitination, suggesting impaired protein turnover. We show cardiac BAG3 operates via chaperone-assisted selective autophagy (CASA), conserved from skeletal muscle, and confirm sarcomeric CASA complex localization is BAG3/proteotoxic stress-dependent. Using mass spectrometry, we characterize the myofilament CASA interactome in the human heart and identify eight clients of BAG3-mediated turnover. To determine if increasing BAG3 expression in HF can restore sarcomere proteostasis/$F_{max}$, HF mice were treated with rAAV9-BAG3. Gene therapy fully rescued $F_{max}$ and CASA protein turnover after four weeks. Our findings indicate BAG3-mediated sarcomere turnover is fundamental for myofilament functional maintenance.

[1] Department of Cell and Molecular Physiology, Loyola University Stritch School of Medicine, Maywood, IL, USA. [2] Department of Medicine, Temple University Lewis Katz School of Medicine, Philadelphia, PA, USA. [3] Department of Medicine, Cleveland Clinic Lerner College of Medicine, Cleveland, OH, USA. [4] Department of Pathology and Laboratory Medicine, Indiana University School of Medicine, Indianapolis, IN, USA. ✉email: jkirk2@luc.edu

Heart failure is the leading cause of morbidity and mortality in the industrialized world and is characterized by impaired contractility and decreased cardiac output[1]. At the cellular level, several factors have been implicated in the development of heart failure, including defective calcium handling, neurohormonal imbalance, and functional decline of the sarcomere—the fundamental molecular unit of contraction[2–4]. The sarcomere is a highly organized protein complex that mediates contraction through myosin and actin filaments, which engage in the calcium-dependent crossbridge cycle. Increased myofilament calcium sensitivity and decreased maximum force-generating capacity ($F_{max}$) are well-documented in heart failure[5–10]. However, while altered calcium sensitivity is uniformly attributed to changes in site-specific post-translational modifications, most commonly phosphorylation of troponin I[11], the molecular mechanisms for decreased $F_{max}$ are incompletely understood.

The sarcomere is under constant mechanical strain, which predisposes its members to denaturation, and yet must maintain optimal mechanical function for decades in the terminally differentiated adult cardiomyocyte. To preserve function, stress-denatured sarcomere proteins must be refolded or targeted to degradation pathways by molecular chaperones so that new proteins can be incorporated[12]. However, sarcomeric protein quality control (PQC) is frequently impaired in heart failure, leading to toxic aggregation of misfolded/dysfunctional proteins[12]. An arrested removal of dysfunctional sarcomere proteins in heart failure may thus serve as a plausible mechanism for the observed decrease in $F_{max}$. However, the distinct mechanisms of sarcomere protein turnover in the adult heart are poorly characterized, and the functional relevance of sarcomere PQC has not been determined.

Bcl2-associated athanogene 3 (BAG3) is a heat shock protein (HSP) co-chaperone that mediates protein degradation through autophagy[13]. Evidence in recent years indicates an essential role for BAG3 in the heart with numerous clinical studies finding that BAG3 mutations are associated with dilated cardiomyopathy (DCM) and myofibrillar myopathy[14–20]. Due to limitations with long-term culture of adult cardiomyocytes, mechanistic studies of BAG3 have primarily been performed in immature neonatal myocytes, where BAG3 was shown to stabilize sarcomere structure through maintenance of the actin-capping protein CapZβ[21]. However, the role of BAG3 at the sarcomere in adult myocytes under mechanical load is less understood and the association with CapZβ does not appear to be conserved[22]. Moreover, sarcomere structure is unimpaired at the neonatal stage with germline BAG3 knockout (KO), but rapidly disintegrates postnatum[23]. These findings suggest separate or additional sarcomeric roles for BAG3 in the adult.

An important study of BAG3 in the adult heart came from Fang et al. who showed BAG3 KO in mice caused DCM, destabilized the small HSPs (HSPBs), and resulted in protein aggregation, indicating impaired protein turnover[24]. Notably, BAG3 KO in iPSC-derived cardiomyocytes caused reduced contractility and altered expression of HSPs, suggesting sarcomeric functional relevance for BAG3-dependent PQC[25]. However, iPSC-cardiomyocytes also have immature morphology and are not under mechanical load, making functional inferences to the adult problematic, especially at the sarcomere, which only fully matures under chronic load. While little is known regarding BAG3's role in the mature cardiac sarcomere, studies in mature skeletal muscle identified BAG3 assembles with HSP70 and HSPB8, forming the chaperone-assisted selective autophagy (CASA) complex, which mediates the turnover of the actin crosslinking protein filamin-C[26–28]. Whether CASA operates at the cardiac sarcomere, the myofilament functional impact of BAG3, and the

**Table 1 Clinical characteristics of the human heart tissue samples.**

| Characteristic | Nonfailing ($n = 9$) | DCM ($n = 21$) | P-value |
|---|---|---|---|
| Age, (years), Mean ± SD | 53.3 ± 8.9 | 55.5 ± 11.5 | 0.66 |
| Female, n, (%) | 4 (44.4) | 8 (38.1) | |
| Ethnicity, n, (%) | | | |
| White | 8 (88.9) | 16 (76.2) | |
| Black | 1 (11.1) | 4 (19.1) | |
| Hispanic | 0 (0.0) | 1 (4.8) | |
| LVEF, (%), Mean ± SD | 57.5 ± 3.2 | 20.3 ± 2.2 | <0.001 |

effect of heart failure on these mechanisms are unknown. However, given the near universally observed depression of $F_{max}$ in systolic heart failure, sarcomeric PQC represents an attractive therapeutic target.

Here, we show that sarcomere protein turnover is impaired in human heart failure samples with reduced $F_{max}$. We verify the presence of the CASA complex at the sarcomere in the human heart, confirm its role in filamin-C turnover therein, and identify seven novel clients of BAG3-mediated protein turnover. We further show myofilament BAG3 expression decreases in human DCM and correlates with functional decline, where lowest BAG3 expression corresponds to weakest force generation. BAG3 haploinsufficient mice also exhibit this functional deficit and display increased myofilament protein ubiquitination and reduced CASA protein expression. Finally, increasing BAG3 using adeno-associated virus gene therapy in mice with heart failure restores myofilament $F_{max}$ and sarcomere protein turnover. Our findings indicate that impaired sarcomere protein turnover contributes to decreased $F_{max}$ in heart failure and highlight BAG3 as an essential factor for sarcomere functional maintenance.

## Results

**Human DCM cardiomyocytes have reduced sarcomere contractile function and impaired myofilament protein turnover.** Decreased contractility in heart failure has been linked to sarcomere dysfunction[3]. In cases of genetic heart failure, decreased $F_{max}$ may be explained by structural alterations to proteins involved in the crossbridge cycle, which are detrimental to their function[29,30]. However, functional deficits have also been observed in nongenetic heart failure[6,7,9,29], which makes up the majority of cases. We therefore sought to determine the underlying mechanism for myofilament dysfunction in heart failure using left ventricle (LV) samples from patients with idiopathic dilated cardiomyopathy (DCM) and heart failure (Table 1).

Using skinned myocytes from DCM and nonfailing (NF) samples, we first showed that myofilament $F_{max}$ decreased significantly in DCM (Fig. 1a, b). Consistent with prior studies discussed above, the $EC_{50}$ (calcium concentration required to elicit half maximal force) also decreased in DCM, indicating increased myofilament calcium sensitivity (Fig. 1c). Since impairment to PQC mechanisms is a common feature of heart failure[31], we hypothesized that disrupted removal of misfolded/dysfunctional sarcomere proteins might underly the observed decrease in $F_{max}$. With functioning PQC, misfolded proteins are tagged with ubiquitin and then removed through various protein degradation mechanisms[12]. Thus, increased ubiquitin levels may serve as a readout for disrupted protein turnover, where proteins are effectively marked for degradation but fail to be turned over. To determine if sarcomere protein turnover was impaired, we assessed ubiquitination in myofilament-enriched fractions from the DCM and NF tissue by western blot. As expected, we found significantly increased ubiquitin at the myofilament in DCM,

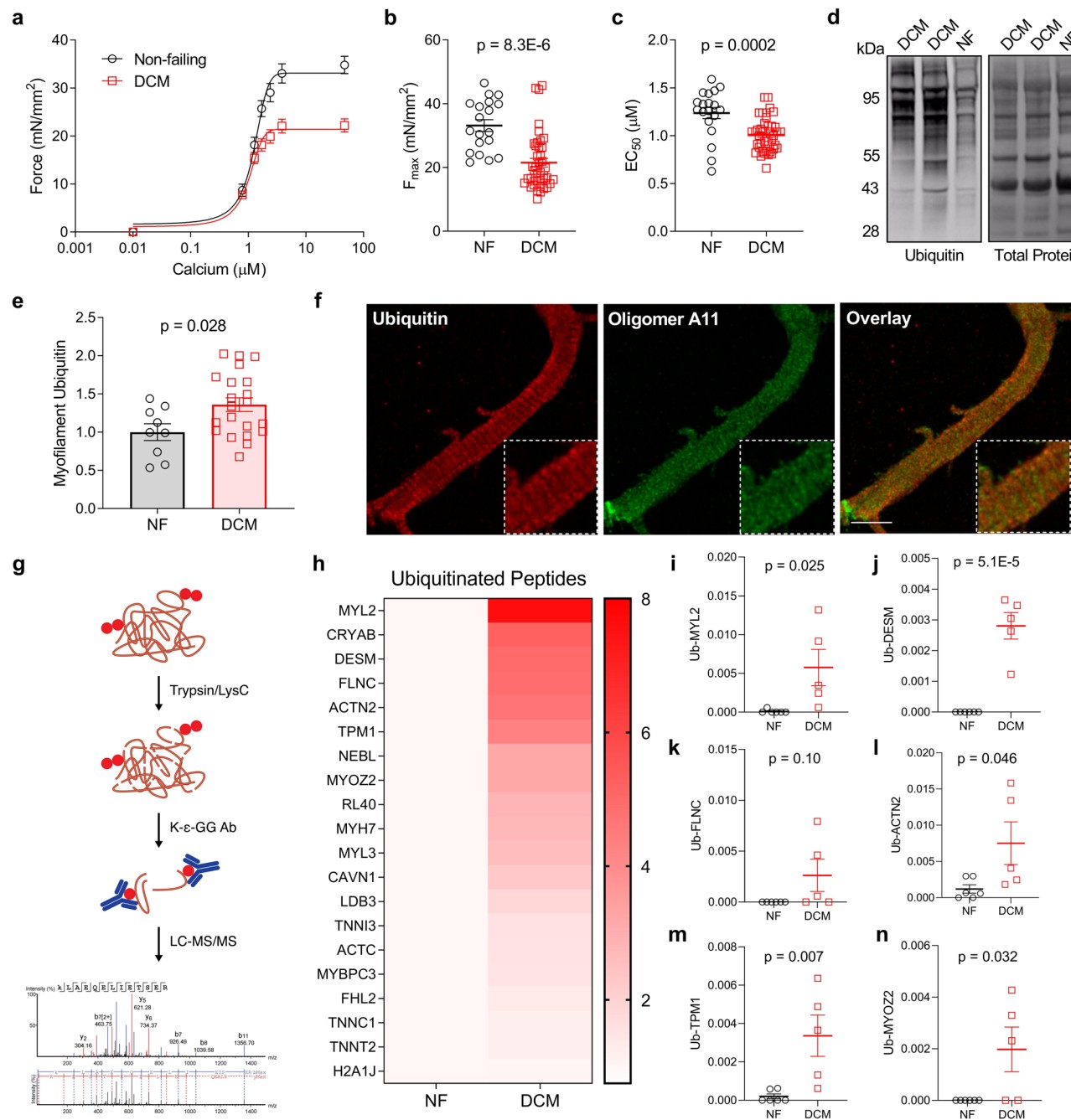

**Fig. 1 Human DCM is characterized by decreased myofilament function and impaired sarcomere protein turnover. a** Skinned myocyte force-calcium relationship from human nonfailing (NF) and dilated cardiomyopathy (DCM) cardiomyocytes; $n = 19$ NF from 6 patients, 41 DCM from 12 patients. **b, c** Summary data for myocyte $F_{max}$ (**b**) and $EC_{50}$ (**c**) corresponding to the force-calcium graph in **a**. **d** Western blot for ubiquitin in the NF and DCM left ventricle (LV) myofilament fraction; image is representative of 9 NF samples and 21 DCM samples. **e** Ubiquitin signal normalized to total protein; $n = 9$ NF, 21 DCM. **f** Immunofluorescence image of a human LV cardiomyocyte immunostained for ubiquitin and oligomer A11; ×63 magnification, scale bar = 10 μm; image is representative of the 16 images acquired. **g** Proteomics paradigm for ubiquitinated peptide enrichment. **h** Heatmap of the top 20 ubiquitinated proteins identified by LC-MS/MS for NF and DCM human patients normalized to total peptide input; scale bar = fold change increase relative to NF, darkest red color denoting greatest fold change. **i–n** Spectral count data for ubiquitinated peptides normalized to total peptide input for myosin regulatory light chain (**i**), desmin (**j**), filamin-C (**k**), α-actinin (**l**), tropomyosin (**m**), and myozenin-2 (**n**); $n = 6$ NF, 5 DCM. All data are presented as mean ± SEM and were analyzed by two-tailed *t*-test.

suggesting a build-up of misfolded sarcomere proteins in disease (Fig. 1d, e).

There are two possible explanations for the observed increase in myofilament ubiquitination in DCM. One is that these ubiquitinated misfolded proteins were removed from the contractile apparatus, but not degraded properly and thus formed protein aggregates that are commonly observed in heart failure[31]. However, a second possibility, which has been suggested in the skeletal muscle sarcomere during atrophy[32], is that misfolded sarcomeric proteins are not being removed from the contractile apparatus and remain integrated into the sarcomere. The protein stoichiometry of the sarcomere is highly conserved[33], and thus

the continued integration of these old sarcomeric proteins, which have been marked for degradation would block the incorporation of newly synthesized proteins, providing an explanation for the decline in $F_{max}$.

To determine whether these ubiquitinated proteins are imbedded within the sarcomere itself, we used confocal microscopy on human DCM cardiomyocytes immunostained for ubiquitin and the protein aggregate marker oligomer A11. As expected, A11 had diffuse punctate localization with no apparent sarcomere patterning. However, ubiquitin localized in a cross-striated pattern and was enriched in thick, regularly repeating bands representative of the sarcomere A-band (Fig. 1f). Among the most abundant A-band proteins are actin, myosin, and myosin binding protein C, which are fundamental to sarcomere contraction[34]. This indicates that the ubiquitinated proteins identified in the myofilament fraction, which were found by western blot to increase in DCM, are in large part incorporated into the sarcomere and not components of protein aggregates. It further suggests that the primary proteins with inadequate turnover in DCM are those directly involved in tension generation.

Next, to confirm the negative functional impact of misfolded proteins remaining integrated into the sarcomere, we treated NF human skinned cardiomyocytes with heat shock at 43 °C for 3 h to cause protein denaturation. Following the heat shock treatment, we performed force-calcium experiments on these myocytes and found, as expected, that heat shock caused a prominent reduction in $F_{max}$ (Supplementary Fig. 1). Since the treatment was of skinned myocytes, which have no intact cellular signaling pathways, the misfolded proteins denatured by heat shock were not tagged with ubiquitin (Supplementary Fig. 1). Therefore, while we could not from this data make any claims as to the functional implications of ubiquitin modifications themselves, we concluded that protein misfolding alone is sufficient to cause significant functional detriment as we hypothesized.

To determine which specific sarcomere proteins had impaired turnover, we digested proteins from the myofilament fraction isolated from NF and DCM tissue with trypsin, enriched for ubiquitinated peptides by immunoprecipitation for the diglycine-lysine ubiquitin remnant motif (K-ε-GG), and identified the peptides via mass spectrometry (Fig. 1g). This technique provided a characterization of the myofilament ubiquitinome in heart failure and identified several proteins with impaired turnover. The top hits with increased ubiquitination in DCM included myosin regulatory light chain-2, desmin, myozenin-2, α-actinin-2, and tropomyosin alpha-1, which had significantly higher ubiquitination when analyzed by spectral count compared to nonfailing (Fig. 1h–n). The total peptide amounts of these proteins were not different between the NF and DCM groups (Supplementary Fig. 2), which is expected given the specific stoichiometry observed by sarcomere proteins. These data indicate sarcomere protein turnover is disrupted in DCM resulting in increased misfolded proteins incorporated into the sarcomere, which offers a compelling possible explanation for the decrease in $F_{max}$.

**Myofilament expression of the co-chaperone BAG3 decreases in DCM and correlates with $F_{max}$.** Several studies have implicated the heat shock protein co-chaperone BAG3 in sarcomere maintenance[20,21,23,35,36], however, the mechanisms of BAG3-mediated sarcomere maintenance in adult cardiomyocytes are poorly understood. BAG3 is involved in PQC by mediating autophagic clearance of misfolded proteins and has been shown to localize to the sarcomere Z-disc, thus positioning it to mediate sarcomere protein turnover[37,38]. We used western blot to assess BAG3 expression in the myofilament protein fraction of human

LV samples and found BAG3 levels decreased significantly in patients with DCM (Fig. 2a, b). Strikingly, when we compared myofilament BAG3 levels with sarcomere functional parameters in the DCM samples, $F_{max}$ was positively associated with BAG3 expression where samples with reduced BAG3 expression displayed significantly lower $F_{max}$ (Fig. 2c). As expected, there was no association between BAG3 levels and calcium sensitivity (Fig. 2d). Importantly, this relationship appears to be specific to the myofilament pool of BAG3, as the correlation diminished when $F_{max}$ was compared with cytosolic/soluble BAG3 expression (Supplementary Fig. 3).

To confirm the relationship between reduced BAG3 expression and sarcomere dysfunction, we next employed a mouse model with cardiomyocyte-restricted heterozygous BAG3 expression ($BAG3^{+/-}$). These mice were previously found to have progressive LV dysfunction starting at 8 weeks of age[39]. In the present study, we used 6-week-old mice to capture the early effects of BAG3 haploinsufficiency on myofilament function, rather than the general response to heart failure development. At 6 weeks, the mice displayed a ~20% reduction in myofilament BAG3 compared to wild-type, comparable to the reduction in human DCM, and had significantly increased myofilament protein ubiquitination (Fig. 2e–h). Extensive sarcomere structural abnormalities, which have been identified in BAG3 germline KO mice at 3 weeks[23] and in BAG3 KO iPSC-cardiomyocytes[35], were not observed in the $BAG3^{+/-}$ mice at this time-point (Fig. 2i). We next assessed myofilament function using force-calcium experiments on skinned myocytes from the $BAG3^{+/-}$ and WT mice and found cardiomyocytes from $BAG3^{+/-}$ mice had reduced $F_{max}$ but no change in calcium sensitivity (Fig. 2j–l). Together, these data indicate BAG3 is required for optimal contractile function of the sarcomere and for maintaining sarcomere proteostasis. Additionally, sarcomere dysfunction due to BAG3 haploinsufficiency arises despite normal sarcomere morphology, suggesting impaired sarcomere protein turnover—not structural disarray—may be the initial insult that leads to contractile dysfunction and ultimately heart failure. It further agrees with our finding that the ubiquitinated proteins remain integrated in the sarcomere and induce dysfunction but not necessarily disarray.

A particularly penetrant BAG3 mutation, that results from a single amino acid substitution of proline at 209 for leucine (P209L), precludes ubiquitinated substrate targeting to the autophagy pathway and causes protein aggregation and myofibrillar myopathy[20,36,40]. Earlier work found that mice with cardiomyocyte-restricted expression of the human P209L bag3 transgene developed restrictive cardiomyopathy by 8 months of age, which was accompanied by elevated whole LV ubiquitin levels and protein aggregation[41]. We found that protein ubiquitination also increases in the myofilament-specific fraction of the P209L mice (Supplementary Fig. 4). Therefore, we employed this model to directly assess the impact of disrupted BAG3-mediated protein degradation on myofilament function. Using skinned myocyte force-calcium experiments on cardiomyocytes from 8-month-old P209L mice, we found $F_{max}$ was significantly reduced compared with aged-matched littermate controls (Supplementary Fig. 5). Taken together, the relationship between decreased/dysfunctional BAG3 and reduced force-generating capacity identified in the human DCM samples, the $BAG3^{+/-}$ mice, and the P209L mice indicates BAG3 is essential for maintaining sarcomere function.

**The chaperone-assisted selective autophagy (CASA) complex localizes to the sarcomere Z-disc and is upregulated in response to proteotoxic stress.** Having established that BAG3 is functionally significant for the sarcomere, we next sought to

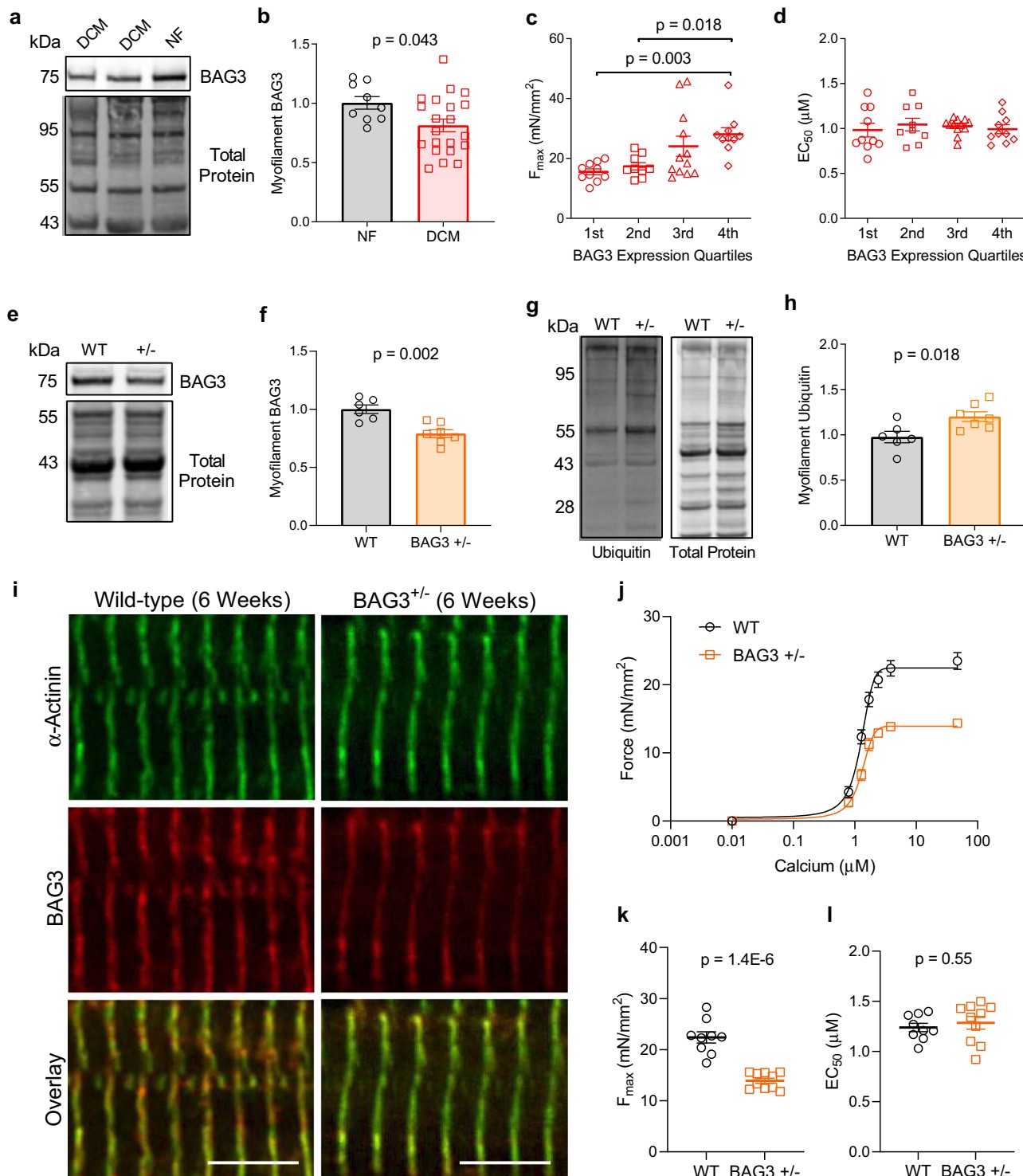

**Fig. 2 Sarcomeric BAG3 expression decreases in DCM and is associated with reduced myofilament $F_{max}$. a** Western blot for myofilament BAG3 in NF and DCM humans; image is representative of 9 NF samples and 21 DCM samples. **b** Myofilament BAG3 signal normalized to total protein; $n = 9$ NF, 21 DCM; two-tailed $t$-test. **c, d** DCM myocyte $F_{max}$ (**c**) and $EC_{50}$ (**d**) grouped by quartile of BAG3 expression; 1st = lowest BAG3 expressors; $n = 12$ DCM samples, 3–4 myocytes/sample for functional assessment; one-way ANOVA, Tukey post-hoc. **e** Western blot for myofilament BAG3 in WT and BAG3$^{+/-}$ mice; image is representative of six WT and seven BAG3$^{+/-}$ samples. **f** BAG3 signal normalized to total protein; $n = 6$ WT, 7 BAG3$^{+/-}$; two-tailed $t$-test. **g** Western blot for myofilament ubiquitin in WT and BAG3$^{+/-}$ mice; image is representative of six WT and seven BAG3$^{+/-}$ samples. **h** Ubiquitin signal normalized to total protein; $n = 6$ WT, 7 BAG3$^{+/-}$; two-tailed $t$-test. **i** Immunofluorescence image of WT and BAG3$^{+/-}$ cardiomyocytes immunostained for α-actinin and BAG3; ×63 magnification, scale bars = 5 μm; image is representative of the 5/group acquired. **j** Skinned myocyte force-calcium relationship from WT and BAG3$^{+/-}$ cardiomyocytes; $n = 9$ WT from 3 mice, 10 BAG3$^{+/-}$ from 3 mice. **k, l** Summary data for myocyte $F_{max}$ (**k**) and $EC_{50}$ (**l**) corresponding to the force-calcium curves in **j**; two-tailed $t$-test. All data are presented as mean ± SEM.

identify the underlying mechanism. To begin, we explored the BAG3 interactome in the human LV myofilament fraction using immunoprecipitation and mass spectrometry. Mass spectrometry analysis identified numerous sarcomere members, which may represent clients of BAG3-mediated protein turnover, as well as two heat shock proteins: HSP70 and HSPB8 (Supplementary Data 1). Mass spectrometry analysis of the HSP70 and HSPB8 myofilament interactomes further confirmed the association of these three proteins (Supplementary Data 1). These two binding partners are noteworthy, as together with BAG3 they were previously shown to engage in chaperone-assisted selective autophagy (CASA) in skeletal muscle[26,27]. In the canonical CASA pathway, the BAG3/HSP70/HSPB8 complex associates with misfolded proteins, which are ubiquitinated by the E3 ligase c-terminus of HSP70 interacting protein (CHIP) and targeted to the autophagosome for degradation[28]. To confirm the presence of the CASA complex at the cardiac sarcomere, we used immuno-fluorescence microscopy on human LV cardiomyocytes and found BAG3, HSP70, HSPB8, and CHIP each localized to the sarcomere Z-disc, as evident from their co-localization with α-actinin (Fig. 3a, b). We further confirmed this localization was sarcomeric using western blot for the CASA members in human whole LV, soluble, and myofilament fractions (Fig. 3c) and confirmed their association therein using co-immunoprecipitation (Fig. 3d, e). These data confirm that the CASA complex is present at the cardiac sarcomere in humans.

Protein chaperones and co-chaperones are regulated by cell stress, advanced levels of which stimulate increased chaperone protein expression. This was previously shown for BAG3 in a fibroblast-like cell line, where treatment with the proteasome inhibitor MG132 caused an upregulation of BAG3 expression[42]. To determine if the CASA complex exhibited proteotoxic stress-dependent localization to the sarcomere, we isolated neonatal rat ventricular myocytes (NRVMs) and maintained them in culture. To cause proteotoxic stress, the NRVMs were treated with 2 μM MG132 for 24 h. MG132 treatment caused a pronounced upregulation of BAG3, HSP70, HSPB8, and CHIP expression in the myofilament fraction, indicating that CASA localizes to the sarcomere in response to proteotoxic stress (Fig. 4a–g). This finding was substantiated further by confocal imaging of DMSO and MG132-treated NRVMs, which showed BAG3 localization to the Z-disc increased in response to MG132 (Fig. 4h–k).

**HSPB8 and CHIP display BAG3-dependent association with the myofilament.** We next asked whether BAG3 was required for localization of the other CASA members to the myofilament. To test this, we used mouse models of cardiomyocyte-specific heterozygous and homozygous BAG3 deletion. Using western blot in myofilament-enriched LV tissue from wild-type, BAG3$^{+/-}$, and BAG3$^{-/-}$ mice, we found myofilament levels of HSPB8 and CHIP were reduced in the partial and complete absence of BAG3 (Fig. 5a–d). However, myofilament HSP70 expression was not impacted by decreased BAG3, suggesting it requires different mechanism for targeting to the sarcomere. This finding is not surprising as HSP70 is a general chaperone for numerous protein clients and operates in many arms of cellular protein folding and degradation[43]. Nevertheless, that myofilament CHIP was reduced with decreased BAG3 is significant for HSP70-mediated degradation processes as CHIP is the primary E3 ligase for HSP70 clients. However, it should be noted that other E3 ligases have been shown to mediate ubiquitination of HSP70 substrates[44,45], and our mass spectrometry analysis of the CASA protein myofilament interactomes identified the E3 ligases MYCBP2, MIB2, and BRE1A (Supplementary Data 1). We expect these ubiquitin ligases and others contribute to the increased myofilament

ubiquitination we observed in the BAG3$^{+/-}$ mice despite the reduction in CHIP.

Since myofilament BAG3 expression decreases in human DCM, we next sought to determine whether HSPB8 and CHIP expression levels display a similar relationship to BAG3 in humans. The BAG3$^{+/-}$ mice had ~80% of the wild-type myofilament BAG3, which we found to be a functionally significant decrease in expression. Therefore, we separated the human samples into two groups: those with "high" BAG3 expression (≥80% of the mean NF BAG3) and those with "low" BAG3 expression (<80% of the mean NF BAG3). As in the mice, HSP70 did not display BAG3-dependent localization to the myofilament (Fig. 5e, f). However, patients with lowest BAG3 expression had significantly reduced myofilament HSPB8 levels (Fig. 5g, h). Myofilament CHIP expression was not significantly decreased in the low BAG3 expressors (Fig. 5i, j). Our data in the BAG3-deficient mice indicate BAG3 is required for assembly of HSPB8 and CHIP at the sarcomere, which is further supported for HSPB8 in the human comparisons. These effects may be through directly mediating HSPB8/CHIP localization to the myofilament or stabilizing these proteins to prevent their degradation, which is supported for HSPB8 by a previous study[24].

**Eight sarcomere proteins identified as candidates for BAG3/CASA-mediated turnover.** Now that we had confirmed the CASA complex localizes to the sarcomere and does so in response to proteotoxic stress, we next sought to identify the sarcomere proteins that are degraded by this pathway. To begin we analyzed the myofilament interactomes of BAG3, HSP70, and HSPB8 identified earlier (Fig. 3) anticipating that CASA clients would be included in the interactome of each CASA member. We identified 52 proteins by this approach, the three primary CASA members and 49 potential clients, that were included in each protein's interactome (Fig. 6a, b). These potential CASA client candidates were primarily sarcomere constituents, with a large proportion having roles at the thin filament (Fig. 6b). While this approach does not ensure that these proteins are CASA clients, it does help to narrow the pool of potential substrates.

As discussed above, previous work in skeletal muscle showed that the actin crosslinking protein filamin-C is degraded by CASA when it becomes denatured due to mechanical strain. The initial characterization of CASA was by Arndt et al. who showed filamin-C was a client by incubating the cytoskeletal fraction from C2C12 skeletal myocytes with recombinant BAG3, which resulted in filamin-C being released to the soluble fraction as identified by western blot[26]. We modified this approach to identify the CASA clients in the heart, and incubated myofilament fractions from human DCM samples with 300 nM recombinant BAG3 for 1 h. We then collected the released/soluble and myofilament/insoluble fractions. Western blot analysis of these fractions indicated that ubiquitinated proteins were released from the myofilament in response to BAG3 treatment (Fig. 6c–e). To identify which specific proteins were released from the myofilament with increased BAG3, we proceeded to trypsin digestion upstream of peptide identification by bottom-up mass spectrometry. Mass spectrometry analysis revealed eight sarcomere proteins that increased in the released pool with BAG3 treatment compared with untreated (Fig. 6f–m). Our findings confirm filamin-C as a client of BAG3-mediated turnover in the human heart and identify seven new candidates: α-actinin, desmin, myosin binding protein C, myomesin-1, myozenin-2, spectrin alpha, and XIRP1.

**BAG3 gene therapy in heart failure restores myofilament F$_{max}$ and CASA protein turnover.** Having established BAG3 is critical for maintaining sarcomere function, identifying CASA as the

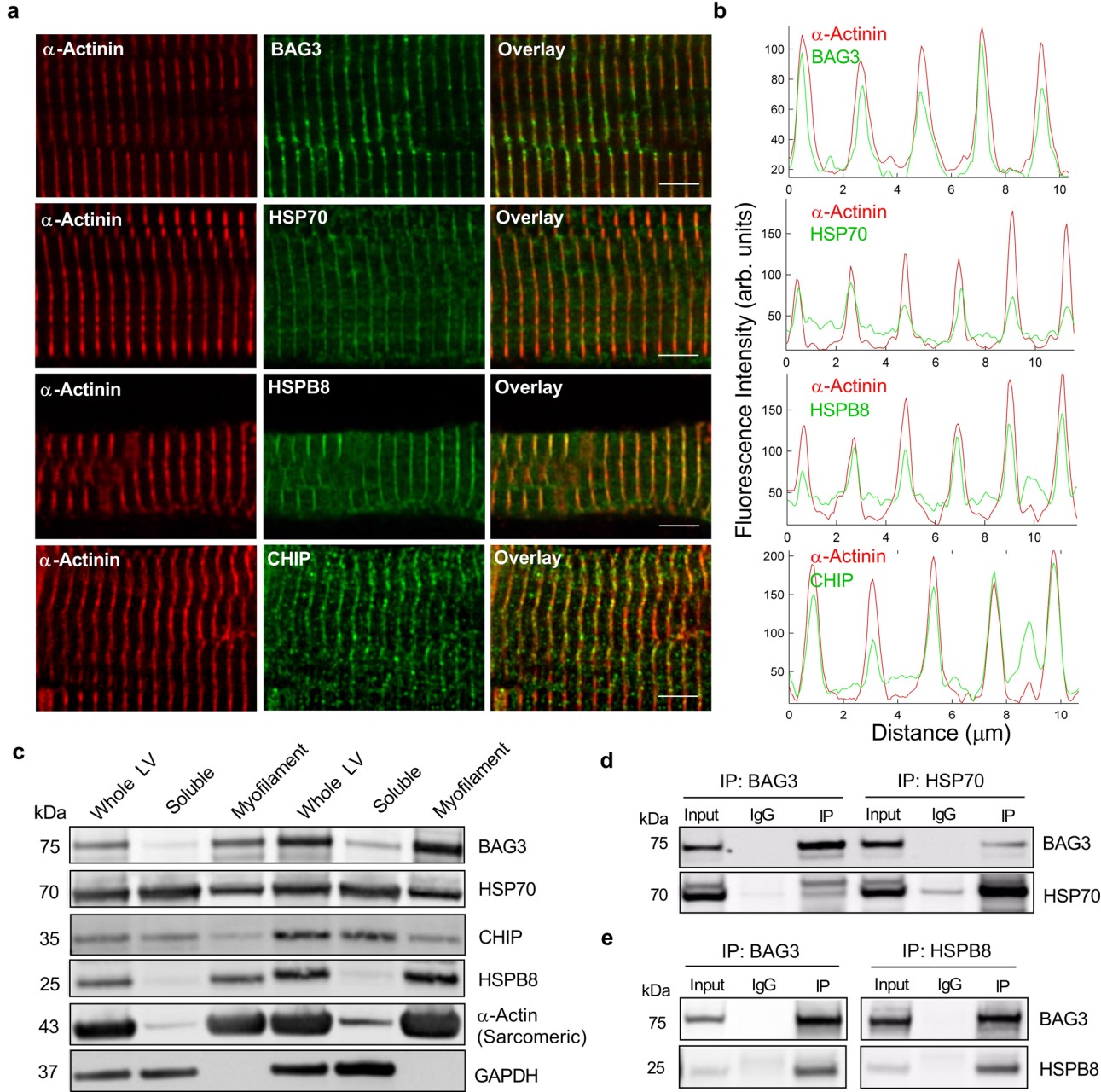

**Fig. 3 The CASA complex localizes to the sarcomere Z-disc in cardiomyocytes. a** Immunofluorescence images of human LV cardiomyocytes immunostained for BAG3, HSP70, HSPB8, and CHIP and counterstained for α-actinin; ×63 magnification, scale bars = 5 μm; images are representative 10 images/antibody/5 independent biological samples. **b** Quantitative line scan of fluorescence intensity by distance from the corresponding green and red channels in panel **a**. **c** Western blot for the CASA complex proteins in the whole LV, triton-soluble, and myofilament fractions; sarcomeric α-actin = myofilament fraction positive control, GAPDH = soluble fraction positive control. **d** Western blot showing results of reciprocal co-immunoprecipitation (IP) for BAG3 and HSP70 in myofilament fraction (IgG—non--specific antibody control); images are representative of three independent experiments. **e** Western blot showing results of reciprocal co-immunoprecipitation for BAG3 and HSPB8 in myofilament fraction; images are representative of three independent experiments.

mechanism, and showing that increasing BAG3 levels causes release of ubiquitinated proteins from the sarcomere, we next hypothesized that increasing BAG3 expression in heart failure would rescue $F_{max}$ through restored CASA. Eight-week-old mice received either a myocardial infarction produced by permanent coronary artery ligation (HF) or sham surgery (Sham). Eight weeks post-surgery, the mice were randomly assigned to receive a recombinant adeno-associated virus serotype 9 (rAAV9) vector expressing the mouse *bag3* gene or control (rAAV9/GFP) via retro-orbital injection. After 4 weeks of adenovirus expression the

mice were euthanized, and the LV tissue was collected. BAG3 overexpression in heart failure restored in vivo cardiac function, which was reported previously for this cohort[46].

We assessed myofilament function using force-calcium experiments on skinned myocytes from the infarct border zone. As in human DCM, cardiomyocytes from mice in the HF group displayed significantly reduced $F_{max}$. However, $F_{max}$ was fully rescued in the HF mice that received rAAV9/BAG3 (Fig. 7a, b). Changes in myofilament calcium sensitivity were not observed, which may be attributed to the relatively early stage

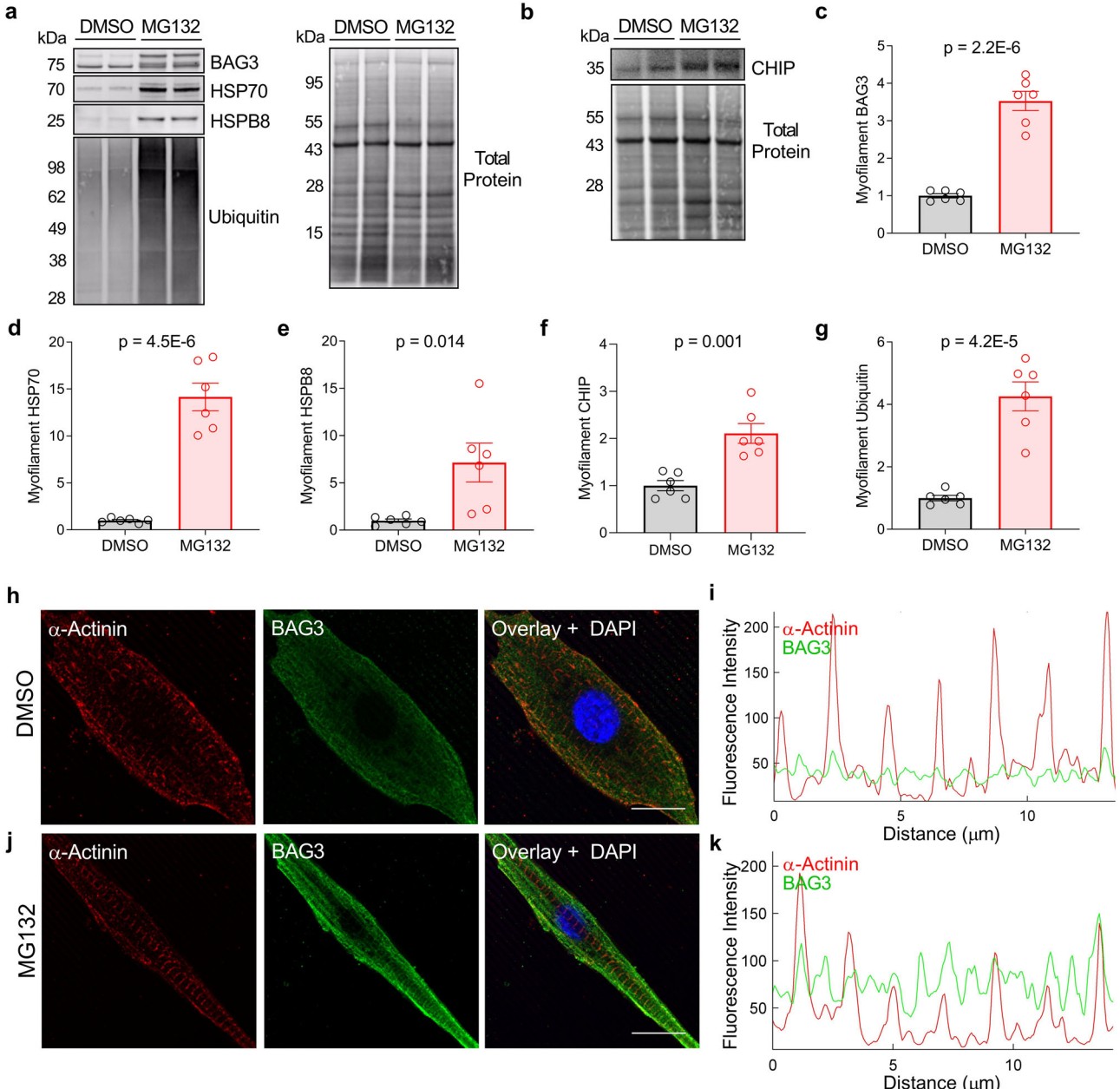

**Fig. 4 The CASA complex is targeted to the sarcomere in response to proteotoxic stress. a, b** Western blots for myofilament BAG3, HSP70, HSPB8, ubiquitin, and CHIP in neonatal rat ventricular myocytes (NRVMs) treated with vehicle (Dimethyl sulfoxide—DMSO) or the proteasome inhibitor MG132; images are representative of 3 independent experiments. **c–g** Quantification of myofilament protein expression normalized to total protein for BAG3 (**c**), HSP70 (**d**), HSPB8 (**e**), CHIP (**f**), and ubiquitin (**g**); $n = 6$ DMSO, 6 MG132 from 3 independent experiments; two-tailed $t$-test. All data are presented as mean ± SEM. **h–k** Immunofluorescence images for NRVMs treated with DMSO (**h**) or MG132 (**j**) immunostained for BAG3 and α-actinin, with quantitative line scan of fluorescence intensity (**i, k**); ×63 magnification, scale bars = 10 μm; images are representative of 5 cells/treatment/3 independent experiments.

of disease progression we studied (Fig. 7c). We hypothesized that the rescue of $F_{max}$ with BAG3 overexpression was due to restoration of CASA client clearance. Therefore, we next assessed expression of the CASA members in the myofilament fraction.

Using western blot, we found that HSP70, HSPB8, and CHIP levels each had increased expression at the myofilament in the HF mice (Fig. 7d–k). BAG3 expression also displayed a mild, though nonsignificant increase in the HF group. These results indicate that targeting of the complex to the myofilament in response to proteotoxic stress is not hindered in the early progression to heart failure, whereas in the end-stage of heart failure (human data) the stress-responsiveness of BAG3/CASA becomes dysregulated. At

this stage, the clearance of the CASA complex and its ubiquitinated clients appears to be either stalled or occurring at an inadequate rate for the level of protein misfolding. However, the issue of CASA clearance was ameliorated by BAG3 gene therapy, which returned CASA expression to Sham levels, supporting restored turnover (Fig. 7d–k). This finding agrees with our client release experiment, where increasing BAG3 protein levels resulted in release of HSP70, HSPB8, and eight sarcomere proteins from the myofilament (Fig. 6 and Supplementary Fig. 5). Notably, myofilament levels of the autophagic ubiquitin receptor P62/SQSTM1, an essential adaptor protein for autophagy that mediates the targeting of ubiquitinated proteins to

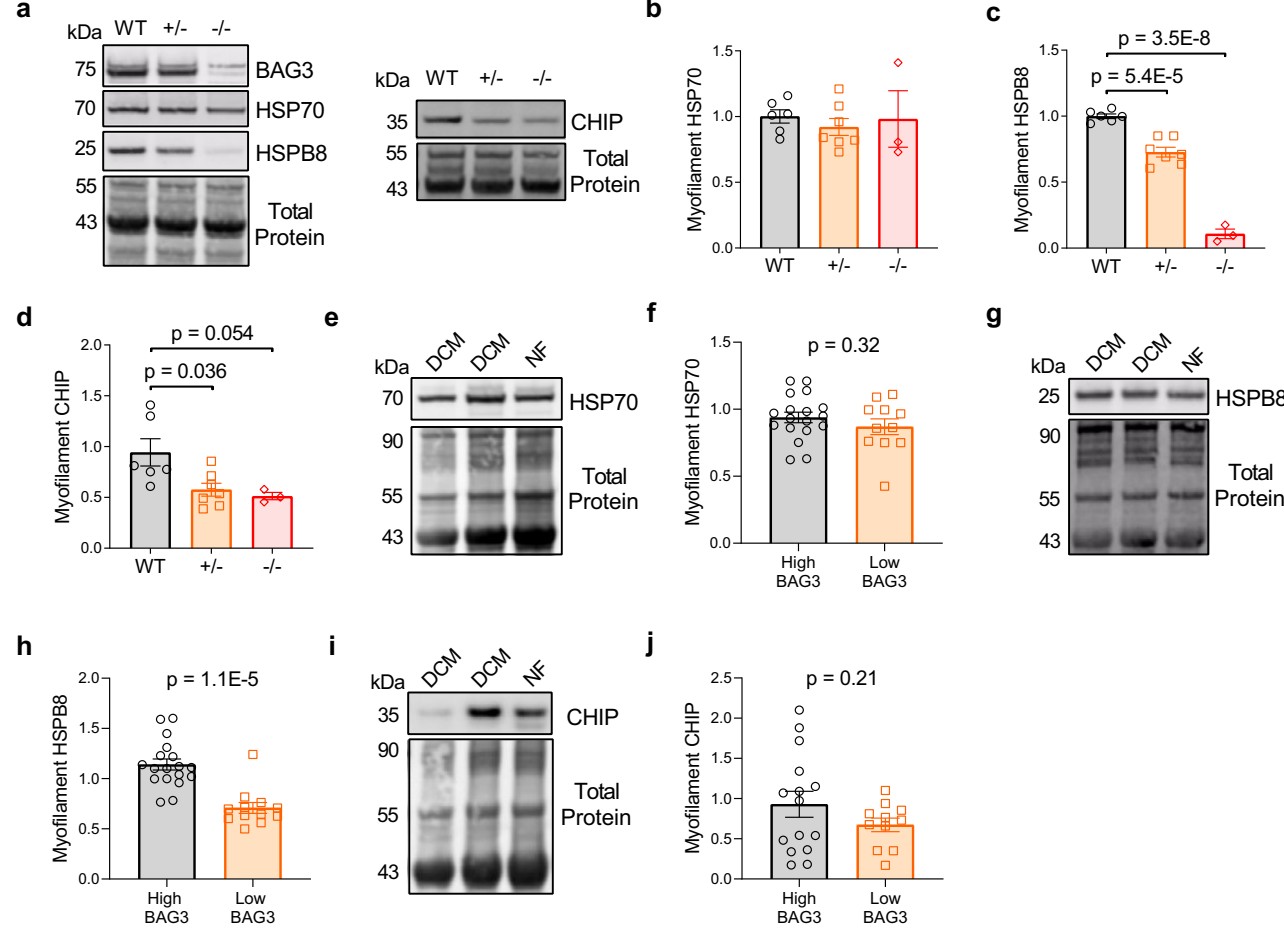

**Fig. 5 BAG3 is required for full assembly of CASA complex members in the myofilament fraction. a** Western blots for myofilament BAG3, HSP70, HSPB8, and CHIP in WT, BAG3$^{+/-}$, and BAG3$^{-/-}$ mice; images are representative of 6 WT hearts, 7 BAG3$^{+/-}$ hearts, and 3 BAG3$^{-/-}$ hearts. **b–d** Densitometry signal of HSP70 (**b**), HSPB8 (**c**), and CHIP (**d**) normalized to total protein; $n = 6$ WT, 7 BAG3$^{+/-}$, 3 BAG3$^{-/-}$; one-way ANOVA, Tukey post-hoc. **e** Western blot for myofilament HSP70 in DCM and NF human samples; image is representative of 9 NF hearts and 21 DCM hearts. **f** HSP70 expression grouped by "high" or "low" BAG3 expression; $n = 18$ high BAG3, 12 low BAG3. **g** Western blot for myofilament HSPB8 in NF and DCM human samples; image is representative of 9 NF hearts and 21 DCM hearts. **h** HSPB8 expression grouped by "high" or "low" BAG3 expression; $n = 18$ high BAG3, 12 low BAG3. **i** Western blot for myofilament CHIP in DCM and NF human samples; image is representative of and 9 NF and 17 DCM hearts. **j** CHIP expression grouped by "high" or "low" BAG3 expression; $n = 15$ high BAG3, 11 low BAG3 (4 samples not assessed due to lack of remaining tissue). High ≥80% of mean NF expression, Low <80% mean NF expression; two-tailed $t$-test. All data are presented as mean ± SEM.

the autophagosome[47], also increased in the HF mice and were returned to baseline with BAG3 gene therapy (Fig. 7j), further indicating restored clearance.

Finally, to determine the impact of BAG3 gene therapy on sarcomere proteostasis, we assessed myofilament protein ubiquitination between the three groups by western blot. As identified in human DCM, we found myofilament protein ubiquitination increased significantly in the HF mice, indicating impaired protein turnover (Fig. 7l, m). However, in the HF mice that received 4 weeks of BAG3 gene therapy, myofilament ubiquitination was significantly reduced (Fig. 7l, m). These data and our preceding data together support that the mechanism by which increasing BAG3 expression in heart failure rescues myofilament $F_{max}$ is through restoring sarcomere protein turnover, where old/misfolded proteins are removed from the structure by CASA and thus allow newly synthesized proteins to be incorporated.

## Discussion

Decreased contractility is a hallmark feature of systolic heart failure. At the cellular level, this may be attributed to structural

and functional changes at the sarcomere, the molecular unit of contraction in striated muscle[3]. However, the mechanisms underlying decreased sarcomere tension generation in heart failure are incompletely understood. In this study, we asked whether inadequate sarcomeric protein turnover in heart failure could serve as an explanation for functional impairment and, if so, to identify the key players involved. We show sarcomere protein turnover is impaired in human DCM samples with reduced myofilament force-generating capacity ($F_{max}$), where elevated ubiquitinated (and thus presumably misfolded/dysfunctional) proteins remained integrated into the sarcomere. We found decreased myofilament localization of the co-chaperone BAG3, such as occurs in heart failure, is directly related to the $F_{max}$ decline and show that increasing BAG3 expression in heart failure restores both function and sarcomere proteostasis through CASA. This study indicates BAG3 is critically important for functional maintenance of the cardiac sarcomere through mediating sarcomere protein turnover.

Disrupted proteostasis has long been associated with sarcomere structural abnormalities and heart failure[12,48–50]. However, mechanisms of sarcomere protein turnover and the myofilament

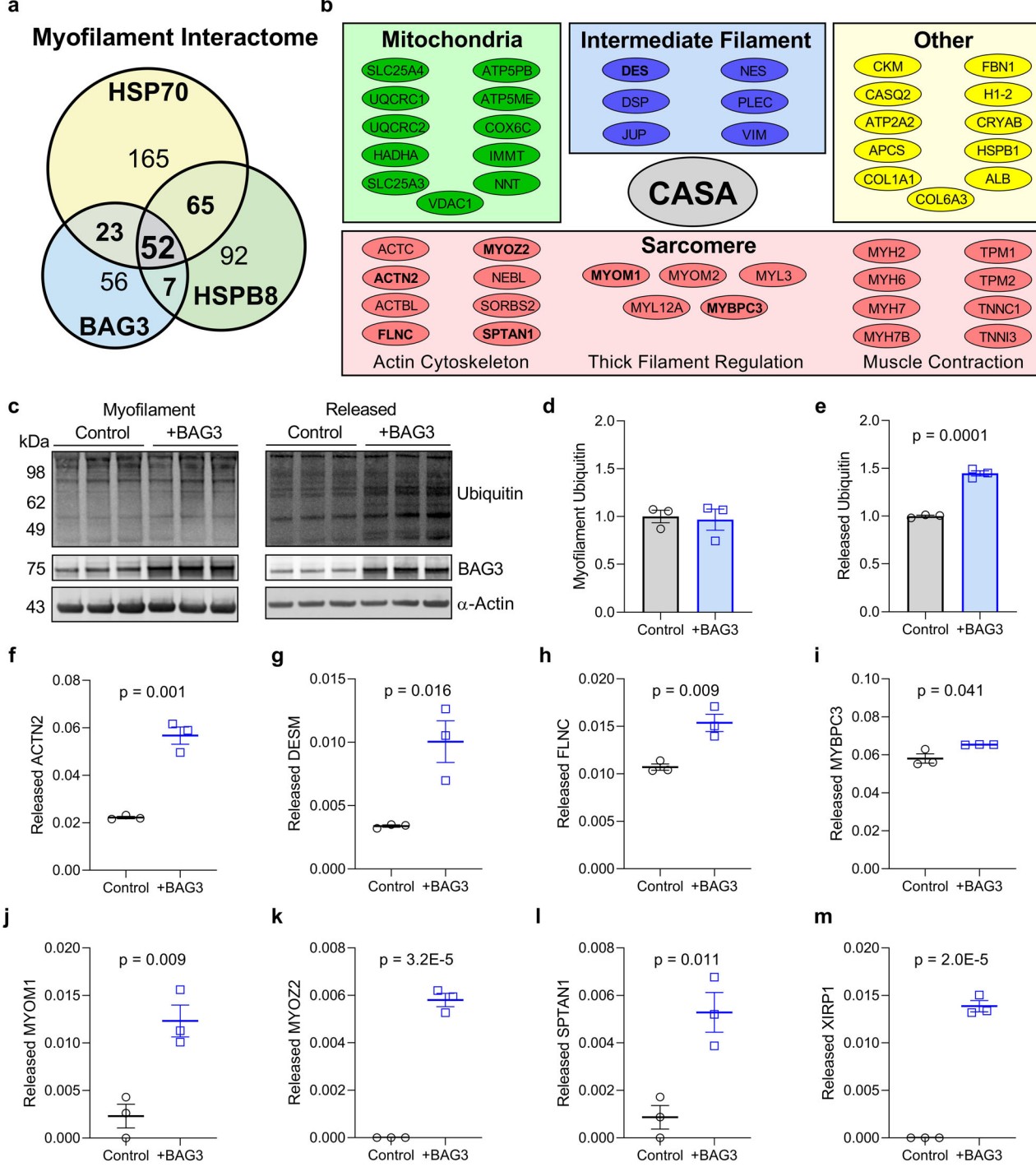

**Fig. 6 Identification of the sarcomeric CASA clients in the human heart. a** Quantitative Venn diagram of the BAG3, HSP70, and HSPB8 myofilament interactomes identified by bottom-up mass spectrometry of immunoprecipitated proteins. **b** List of the 49 shared proteins identified in the myofilament interactome of each CASA complex member; bold = previously identified CASA clients (FLNC) and new CASA clients identified in this study. **c** Western blot for myofilament ubiquitin and ubiquitinated proteins released from the myofilament in response to increased BAG3 expression; image is representative of 3 hearts/group. **d, e** Quantitative densitometry for myofilament (**d**) and released (**e**) ubiquitin normalized to α-actin; n = 3 control, 3 + BAG3. **f–m** Spectral count analysis of the eight sarcomere proteins identified by mass spectrometry to increase in the released protein fraction with increased BAG3 expression. For all, n = 3 control and 3 recombinant BAG3 treatment (+BAG3), from the same 3 human LV samples. All data are presented as mean ± SEM and were analyzed by two-tailed t-test.

functional significance of individual molecular chaperones involved in sarcomere PQC are not known. BAG3 is key to PQC in many different cell types and has been shown to target already ubiquitinated proteins to autophagy and to directly facilitate the ubiquitination of other misfolded proteins, leading to their autophagic clearance[51,52]. Interestingly, BAG3 had previously been shown to localize to the sarcomere Z-disc, positioning it as a potential mediator of sarcomere protein turnover through

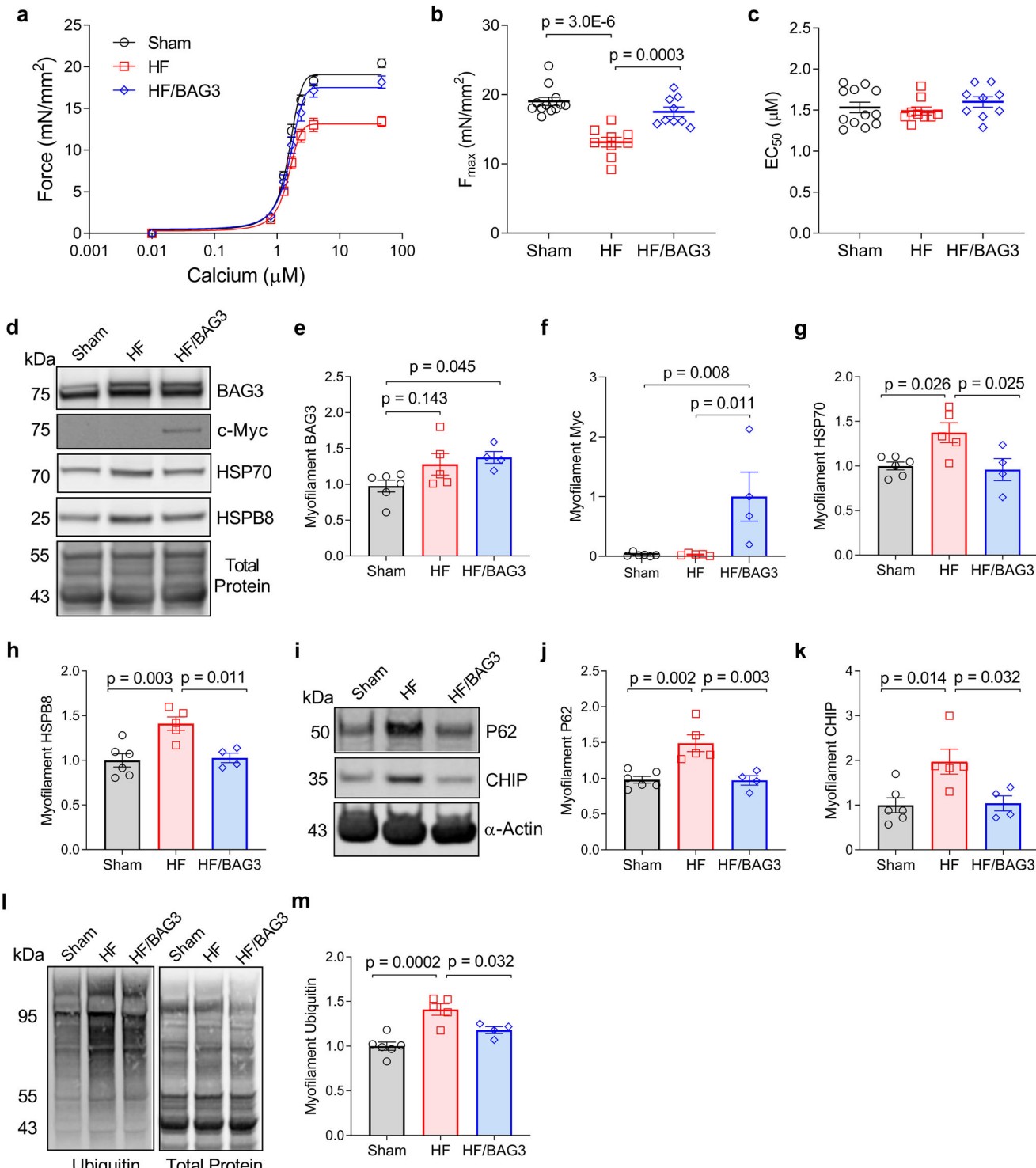

**Fig. 7 Increasing BAG3 expression in heart failure restores myofilament function and sarcomeric CASA turnover. a** Skinned myocyte force-calcium relationship for mouse cardiomyocytes from sham, heart failure (HF), and HF treated with AAV9-BAG3 (HF/BAG3); $n = 12$ sham from 4 mice, 9 HF from 3 mice, and 9 HF/BAG3 from 3 mice. **b, c** Summary data for myocyte $F_{max}$ (**b**) and $EC_{50}$ (**c**) corresponding to the force-calcium curves in **a**. **d** Western blot for myofilament BAG3, *myc*-BAG3, HSP70, and HSPB8 in the sham, HF, and HF/BAG3 mice; images are representative of 6 sham hearts, 5 HF hearts, and 4 HF/BAG3 hearts. **e–h** BAG3, myc, HSP70, and HSPB8 protein expression normalized to total protein. **i** Western blot for myofilament P62 and CHIP in the sham, HF, and HF/BAG3 mice; images are representative of 6 sham hearts, 5 HF hearts, and 4 HF/BAG3 hearts. **j, k** P62 and CHIP protein expression normalized to total protein. **l** Western blot for myofilament ubiquitin in the sham, HF, and HF/BAG3 mice; images are representative of 6 sham hearts, 5 HF hearts, and 4 HF/BAG3 hearts. **m** Myofilament ubiquitin normalized to total protein. For all western blots: $n = 6$ sham, 5 HF, 4 HF/BAG3. All data are presented as mean ± SEM and were analyzed via one-way ANOVA with Tukey post-hoc test for multiple comparisons.

autophagy[38]. However, relatively little was known regarding BAG3's role at the cardiac sarcomere. Mechanistic studies in neonatal myocytes showed BAG3 was required for sarcomere structural organization through stabilization of the actin-capping protein CapZ[21]. However, neonatal myocytes are structurally and functionally distinct from the adult cardiomyocyte and are not under any mechanical load, making inferences from this study to the adult difficult. Indeed, it is apparent that BAG3's role in adult cardiomyocytes differs from the neonatal as BAG3 was found dispensable for sarcomere structural maintenance in utero, but loss of BAG3 caused rapid sarcomere disintegration once the heart was under mechanical load[23]. Moreover, recent studies in the adult heart failed to identify CapZ as a BAG3-interacting protein, suggesting the stabilization of CapZ by BAG3 is specific to sarcomere assembly[22]. Our mass spectrometry analysis of the myofilament BAG3 interactome also failed to identify CapZ as a binding partner, further suggesting this association is specific to development.

An important advance in our understanding of BAG3's role in the adult heart came from Fang et al. who showed that cardiomyocyte-restricted BAG3 KO caused a DCM phenotype in mice, was associated with reduced stability of the HSPBs, and increased toxic protein aggregation[24]. The authors also showed that a DCM-associated BAG3 mutation impaired BAG3 binding to HSP70 and was associated with reduced stability of HSPBs and protein aggregation. Together, these findings strongly indicate BAG3 is required for protein turnover in the adult heart. However, while this study identified prominent protein aggregation with decreased/dysfunctional BAG3, it did not assess whether misfolded proteins remained incorporated into the sarcomere. In the present study, we found that sarcomere-specific expression of BAG3 decreased in human DCM and levels were particularly low in samples with the weakest myofilament force-generating capacity. We therefore hypothesized that impaired BAG3-mediated turnover of sarcomere proteins might contribute to decreased sarcomere contractile function. Using cardiomyocyte-restricted heterozygous deletion of BAG3, we found elevated sarcomere protein ubiquitination, suggesting inadequate turnover, and reduced myofilament force-generating capacity. Importantly, these ubiquitinated proteins were not members of protein aggregates, but were still imbedded within the sarcomere complex and did not cause overt structural disarray, thus offering a compelling explanation for the observed functional deficit.

To further clarify the functional importance of BAG3-mediated protein turnover, we used a transgenic mouse model expressing the human P209L mutation. The P209L BAG3 mutation is highly penetrant and is associated with myofibrillar myopathy and elevated toxic protein aggregation[20,36]. Recent work in cell culture systems identified P209L BAG3 impairs release of ubiquitinated substrates to the macroautophagy system[40,53]. We chose transgenic expression of the human P209L bag3 as a recent study of the analogous mouse mutation (P215L) showed no cardiac phenotype, which was attributed to the pathogenicity of this variant being specific to the human BAG3 isoform[54]. Previous work with the human P209L transgenic mouse found mice developed restrictive cardiomyopathy by 8 months, mimicking the human phenotype[41]. In the present study, we found expression of the P209L BAG3 mutant increased myofilament ubiquitin levels and caused a significant reduction in $F_{max}$ by 8 months of age. Instead of mediating sarcomere stability as in the neonatal myocyte, our data identify a converse role for BAG3 in the adult myocyte, where BAG3 is required for sarcomere protein turnover.

Studies in skeletal muscle previously identified BAG3 mediates chaperone-assisted selective autophagy (CASA), a macroautophagy pathway involving HSP70 and HSPB8[26,27]. In skeletal muscle, CASA mediates the turnover of the actin cross-linking protein filamin-C when it is denatured due to mechanical strain[26,27]. Misfolded filamin-C is bound by HSP70/HSPB8 which form a complex with BAG3. The HSP70-bound misfolded client is then ubiquitinated through the actions of the E3 ubiquitin ligase CHIP, which associates with the HSP70 c-terminus[28]. We found that CASA is conserved in cardiomyocytes and that the complex assembles at the sarcomere in a BAG3- and proteotoxic stress-dependent manner in cardiomyocytes. Using several mass spectrometry-based approaches, we also confirmed that filamin-C is a client of CASA in the human heart and identified seven additional clients of structural and functional significance for the sarcomere. Notably, five of the eight clients identified are known to associate with the thin filament and engage in regulating thin filament structure and dynamics. Thus, CASA emerges as a mechanism of thin filament maintenance.

In the canonical CASA pathway, the client (filamin-C) was thought to be first bound by the BAG3/HSP70/HSPB8 complex and then ubiquitinated by CHIP and degraded[28]. However, our data indicate that BAG3 can assist in the removal of proteins that are already ubiquitinated and suggests that ubiquitination of HSP70/HSPB8 clients in CASA may occur prior to the association of BAG3 with the complex, though BAG3 is fundamental for clearance of the complex/clients. It is also suggested from our data that CHIP is not the only E3 ligase involved in ubiquitinating CASA clients as myofilament ubiquitination increased, despite a 50% reduction in CHIP in the BAG3[+/−] mice. This finding is not surprising as multiple E3 ligases have been identified for HSP70 clients and our proteomics data indicates several other E3 ligases associate with the CASA members.

An earlier study had shown that BAG3 gene therapy restored in vivo cardiac function in a mouse model of heart failure secondary to coronary artery ligation[46]. Employing this same model, we found that BAG3 gene therapy fully rescued myofilament $F_{max}$ and reduced sarcomere ubiquitination. Notably, the stress-responsiveness of CASA at this early stage in heart failure progression was intact as suggested by significant increases in HSP70, HSPB8, and CHIP at the myofilament. This suggests that at this stage of heart failure, either clearance of the CASA complex stalls or occurs at an inadequate rate. Regardless, CASA protein expression returned to baseline after increasing BAG3, indicating that gene therapy rescued clearance of the complex and its clients. This restoration of CASA clearance with increased BAG3 is supported by our client release experiment in the myofilament fraction from DCM patients, where addition of BAG3 resulted in release of HSP70 and HSPB8 to the soluble fraction along with the ubiquitinated proteins.

Under conditions of elevated stress, such as are present at the sarcomere, proteins are predisposed to misfolding. In the healthy state, misfolded proteins are removed from the sarcomere and swapped with functional replacements. However, protein degradation mechanisms are dysregulated in heart failure. Our data identify inadequate sarcomere protein turnover as a mechanism of sarcomere functional decline in heart failure. Moreover, our findings show decreased BAG3-mediated sarcomere protein degradation is a primary contributor to this observed decrease in myofilament contractile function. This study indicates impaired PQC at the sarcomere in heart failure has a direct effect on myofilament function. Furthermore, our results show that BAG3 is required for maintaining sarcomere proteostasis and optimal tension generation in cardiomyocytes and demonstrate that the potential therapeutic benefits of BAG3 gene therapy in heart failure include improving myofilament function by restoring sarcomeric protein turnover.

## Methods

**Human heart tissue procurement**. Human left-ventricular tissue was obtained from the Loyola Cardiovascular Research Institute Biorepository and Cleveland Clinic Biorepository. Tissue from failing human hearts with non-ischemic idiopathic dilated cardiomyopathy and left-ventricular dysfunction was collected either at the time of heart explant or at the time of LVAD implantation. Informed consent was obtained prior to tissue collection. Myocardial tissue from the nonfailing donors with no history of coronary artery disease or heart failure (EF%, 57.5 ± 3.2) was collected postmortem with other organs. All tissue was flash frozen in liquid nitrogen. The patient age and sex distribution were not significantly different between the nonfailing (53.3 ± 8.9 years, 44.4% female) and failing hearts (55.5 ± 11.5 years, 38.1% female).

**Animals**. All animal experiments were conducted according to the standards set by the Loyola University Chicago Health Science Division, Temple University, and Hines VA Institutional Animal Care and Use Committee (IACUC) in adherence to the US National Institutes of Health Guide for the Care and Use of Laboratory Animals. Experimental procedures were approved by the Loyola University Chicago Health Science Division, Temple University, and Hines VA IACUCs, which are AALAC accredited institutions. Both mice and rats were maintained in 12-h light/dark cycles from 6 AM to 6 PM at 65–75 °F/40–50% humidity and had unrestricted access to food and water.

**AAV9-BAG3 overexpression mouse model**. The generation of mice with LV dysfunction secondary to a left coronary artery ligation was described previously[46]. In brief, eight-week old C57/Bl6 mice were randomized to undergo a left descending coronary artery (LAD) ligation or sham surgery[55]. Because only half of the mice in which the artery was ligated were alive at one-week post-surgery, mice were not randomized to treatment group until 1-week post-surgery. Eight weeks post-MI, the now randomized sham-operated mice and post-infarction mice received either gene therapy with a recombinant adeno-associated virus-9 expressing BAG3 (rAAV9-BAG3) or control (AAV9-GFP) by retro-orbital injection. Four weeks later mice were sacrificed, tissue was removed, and flash frozen in liquid nitrogen.

**BAG3 P209L transgenic mouse model**. The generation of the P209L BAG3 mouse strain was described previously[41]. Transgene-positive founders were bred into wild-type C57/Bl6 mice. P209L positive offspring were identified through PCR of isolated tail DNA obtained during weaning using primers for the human bag3 transgene. Wild-type and transgene-positive offspring were pair-housed and aged to 8 months, at which time they were euthanized, and myocardial tissue was collected.

**BAG3+/− and −/− mouse model**. Generation of mice homozygous (−/−) or heterozygous (+/−) for the murine version of the constitutive (c) deletion of BAG3 (cBAG3−/− or cBAG3+/−) has been described in detail elsewhere[39]. Mice lacking a single allele of wild-type BAG3 developed early LV dysfunction by 8–10 weeks of age. Mice in which both alleles of BAG3 were deleted developed LV dysfunction commensurate with that of the heterozygous mice; however, the BAG3−/− died by 12–14 weeks of age with severe LV dysfunction. Hemodynamic data for the BAG3+/− was reported previously[39]. Hemodynamic data for the BAG3−/− was obtained separately. The mice used for data collection in this study were 6 weeks of age, thus preceding the development of detectable in vivo LV dysfunction.

**Neonatal rat ventricular myocyte studies**. Neonatal rat hearts were isolated from 0–1-day-old Sprague-Dawley rat pups (Charles River Laboratories), placed in ice-cold Krebs-Henseleit Buffer (KHB)[56], and rinsed twice. Next, the atria were removed, and the isolated ventricles were digested in 37 °C KHB supplemented with 1 g:100 mL collagenase type-II (Worthington) and 0.05% trypsin with intermittent stirring. The buffer solution was then moved to a tube containing Dulbecco's Modified Eagle's Medium (DMEM, Gibco) supplemented with 10% fetal bovine serum (FBS, Millipore Sigma) to stop the digestion reaction. This process was repeated four times or until the myocardial tissue was fully digested and turned white. Next, the digested ventricular myocyte solution was centrifuged at $140 \times g$ for 10 min and the pelleted cells were resuspended in DMEM with 10% FBS and penicillin/streptomycin, plated, and incubated at 37 °C in a 5% $CO_2$ incubator for 90 min to allow the fibroblasts to stick. After incubation the plates were softly tapped to dislodge cardiomyocytes, the supernatant containing the cardiomyocytes was collected and cultured at 1 million cells per plate on 60 × 15 mm culture dishes coated with 0.1% gelatin and incubated at 37 °C/5% $CO_2$. After 48 h, the media was replaced with fresh media containing 2 µM MG132 (MedChem Express) dissolved in DMSO to induce proteotoxic stress by inhibiting the proteasome, or equal volume of DMSO (Millipore Sigma) vehicle control. After 24 h of treatment the cells were collected and enriched for myofilament proteins.

**Skinned myocyte Force-Ca²⁺ experiments**. Skinned cardiomyocytes were prepared as described previously[57]. Briefly, frozen left-ventricular tissue was placed in

Isolation solution (Supplementary Table 1) containing protease and phosphatase inhibitors (1:100 ratio, Fisher Scientific), 10 mM 1,4-dithiothreitol (DTT, Millipore Sigma), and 0.3% (v/v) Triton X-100 (Millipore Sigma). The tissue was mechanically homogenized in three one-second bursts and left on ice for 20 min to permeabilize. Next, the myocyte solution was pelleted by centrifugation at $120 \times g$ for 2 min and resuspended in Isolation solution without triton. For the heat shock experiment, the myocytes were split into two tubes: one kept at 4 °C and the other placed on a shaking incubator at 43 °C for three h. For all other experiments, the myocytes were kept on ice.

To assess function, myocytes were first attached with UV-curing glue (NOA 61, Thorlabs) to two pins, one attached to a force transducer (AE-801, Kronex) and the other to a high-speed piezo length controller (Thorlabs). Next, the myocyte was perfused with a maximal calcium Activating solution to elicit contraction and then perfusion was switched to 100% Relax solution (Supplementary Table 1) to facilitate relaxation. This process was continued with perfusion of five additional submaximal calcium solutions containing Activating solution mixed with Relax solution. All measurements were conducted at room temperature and performed at a sarcomere length of 2.1 µm as measured by Fast Fourier Transform. Experimenters were blinded to the experimental groups during the collection of these data.

**Myofilament enrichment**. Frozen left-ventricular tissue was placed in 1 mL of a standard rigor buffer (SRB)[58] containing protease and phosphatase inhibitors (1:100 ratio) and 1% Triton X-100. The tissue homogenized with a mechanical homogenizer and left on ice for 10 min. For NRVMs, the cells were suspended in SRB with Triton and left to permeabilize on ice for 20 min without homogenization. Myofilaments were pelleted by centrifugation at $1800 \times g$ for 2 min and the supernatant containing soluble proteins was removed. The myofilament pellet was washed twice in SRB without Triton, resuspended in 9 M Urea (Millipore Sigma), and sonicated. Following sonication, the solution was centrifuged at $12,000 \times g$ for 10 min and the supernatant containing the now solubilized myofilament proteins was collected.

**Immunoblotting**. After isolating the myofilament fraction, protein concentration was determined using a BCA assay (Thermo Fisher). Samples were prepared in SDS Tris-Glycine Buffer (Life Technologies) supplemented with Bolt Reducing Buffer (Fisher Scientific), heated for 10 min at 95 °C, and run on 4–12% gradient Tris-glycine gels (Invitrogen). The proteins were transferred onto a nitrocellulose membrane (Thermo Scientific). Following transfer, membranes were incubated with Revert Total Protein Stain (LI-COR Biosciences) for 5 min to assess equal loading and then blocked in a 1:1 ratio Intercept Blocking Buffer (LI-COR Biosciences) to Tris-Buffered Saline (TBS) solution for 1 h at room temperature. The following primary antibodies and dilutions were used and incubated in blocking buffer with 0.1% Tween overnight at 4 °C with gentle rocking: BAG3 (Proteintech, 10599-1AP, 1:5,000), BAG3 (Santa Cruz Biotechnology, sc-136467, 1:500), c-Myc (Cell Signaling Technology, 71D10, 1:1000), HSPB8 (Proteintech, 15287-1AP, 1:3,000), HSP70 (Proteintech, 10995-1AP, 1:3,000), Ubiquitin (Cytoskeleton Inc., AUB01, 1:400), CHIP/STUB1 (Santa Cruz Biotechnology, sc-133083, 1:100), P62 (Proteintech, 18420-1AP, 1:3,000), GAPDH (Cell Signaling Technology, 14C10, 1:4000), Sarcomeric α-actin (Millipore Sigma, A2172, 1:10000). Blots were analyzed using the LI-COR Image Studio software.

**Co-immunoprecipitation**. Dynabeads Protein G (Invitrogen) were incubated with 5 µg of antibody in 200 µl TBS with 0.1% Tween (TBS-T) for 20 min at room temperature with rotation. The immunocomplex was washed three times with 100 mM sodium borate (pH 9) and then antibody was crosslinked with 20 mM dimethyl pimelimidate dihydrochloride (DMP) in 100 mM sodium borate for 30 min at room temperature. After crosslinking, the complex was blocked with 200 mM ethanolamine (pH 8) for 2 h to limit nonspecific protein binding. After blocking, the immunocomplex was washed three times with IAP buffer (50 mM MOPS (pH 7.2), 10 mM sodium phosphate, 50 mM NaCl) and then resuspended in 200 µl of 2 µg/µl myofilament-enriched protein lysates. After incubation for 10 min at room temperature with rotation, the immunocomplex was washed seven times with TBS-T on a magnetic rack. Immunoprecipitated proteins were eluted by suspending the beads in a 0.15% TFA solution for 5 min at room temperature. The eluted proteins were dried by vacuum centrifugation and then prepared for mass spectrometry or analyzed by western blot.

**Client release experiment**. This protocol was adapted from Arndt et al. with minor modifications[26]. Approximately 500 µg of myofilament-enriched protein from three separate human DCM LV samples was split into two equal pools and each resuspended in 100 µl of a 20 mM MOPS, 100 mM KCl, 2 mM ATP, 2 mM MgCl₂ buffer. Recombinant BAG3 protein (Abcam) was then added to half of the samples at a final concentration of 300 nM, while the remaining half were left untreated. All samples were then incubated at 30 °C for 1 h with constant agitation. After incubation, the myofilaments were pelleted by centrifugation at $10,000 \times g$ for 10 min. The soluble protein fraction containing the "released" proteins was collected and the myofilament/insoluble protein pellet was solubilized in 500 µl of 9 M urea. To assess ubiquitin in the two fractions, 10 µl of the released protein and

2.5 μl of the myofilament protein were used for western blot. The remaining sample underwent in-solution digestion for mass spectrometry as detailed below. The resulting peptides from the released proteins were reconstituted in 20 μl of buffer A (3% acetonitrile, 0.1% formic acid) and those from the myofilament pellet were reconstituted in 2 mL buffer A. For the analysis, this 100-fold difference in the loading amount was taken into account to allow for appropriate comparison between the fractions.

**In-solution digestion mass spectrometry.** Following immunoprecipitation for BAG3 the co-immunoprecipitated proteins were reconstituted in 50 mM Tris-HCl (pH 8) and reduced with 5 mM DTT for 45 min. Following reduction, the proteins were alkylated by incubation in 10 mM iodoacetamide for 30 min. The proteins were next digested with 5 ug Trypsin/LysC protease for 18 h at 37 °C. The reaction was stopped by adding TFA at 0.1% final volume to bring the pH under 3. Samples were then dried by vacuum centrifugation, reconstituted in 300 μl of 0.1% TFA, and separated into 12 fractions by basic pH reverse phase fractionation. The fractions were concatenated, dried, reconstituted in buffer A (3% acetonitrile, 0.1% formic acid) and analyzed by liquid chromatography tandem mass spectrometry (LC-MS/MS).

**Ubiquitinated myofilament peptide enrichment mass spectrometry.** This procedure was adopted from Udeshi et al.[59] with minor modifications. Approximately 3 mg human or 500 μg of mouse myofilament-enriched protein was incubated in 5 mM DTT at room temperature for 45 min to reduce disulfide bonds, followed by alkylation via incubation with 10 mM IAA for 30 min. The protein was then diluted 1:4 with 50 mM Tris-HCl (pH 9) to reduce the urea concentration and digested with Trypsin/LysC protease mix (1:50 wt/wt) for 18 h at 37 °C (pH 8–8.5). The peptides were next desalted with a 100-mg tC18 SepPak cartridge (Waters), dried by vacuum centrifugation, and stored at −80 °C until next use. At this stage a small amount of sample was set aside to be run as a total peptides control on LC-MS/MS later. Dried peptides were reconstituted in 0.1% TFA and separated into 12 fractions via high pH reverse phase fractionation (Pierce). The collected fractions were pooled by concatenation to maximize LC gradient utilization and then dried by vacuum centrifugation until next use.

Ubiquitinated peptides were purified using immunoaffinity purification with the diglycine-lysine ubiquitin remnant motif antibody (Cell Signaling Technology, PTMScan) crosslinked to Protein A agarose beads. Each pooled fraction was reconstituted in ice-cold IAP buffer (50 mM MOPS pH 7.2, 10 mM NaPO$_4$ dibasic, 50 mM NaCl) and incubated with 30 μg of antibody for 1 h at 4 °C with gentle rotation. After washing, the purified peptides were eluted by incubation with 0.15% TFA and concentrated using the GL-Tip SDB 200 μl C18 tips (GL Biosciences). The collected samples were then dried by vacuum centrifugation and stored at −80 °C until ready for LC-MS/MS.

Tryptic peptides were reconstituted in 20 μl of buffer A (3% ACN, 0.1% FA) and 5 μl were subjected to high pressure liquid chromatography (HPLC) in a 25 cm PepMap RSLC C18 column (Thermofisher) coupled to tandem mass spectrometry (MS/MS) on an LTQ Orbitrap XL mass spectrometer. Upon completion, raw data files were imported into the Peaks Bioinformatics program and acquired masses were searched against the *Homo sapiens* or *Mus musculus* database. The precursor ion mass tolerance was set at 20 ppm and the MS/MS was set at 1.0 Da. A maximum of three missed cleavages were allowed. The fixed modification was carbamidomethylated cysteine (+57.02 Da) and the variable modification was ubiquitinated lysine (+114.04 Da). For the analysis of the human data, the Peaks filtered, database-matched peptides were used. For the mice, the Peaks de novo sequencing peptide data were used. For comparison between groups, ubiquitinated peptides or de novo peptides were normalized to the corresponding total peptides control for each sample set aside earlier.

**Immunofluorescence microscopy.** Frozen left-ventricular tissue was placed in Isolation solution (Supplementary Table 1) containing 0.3% Triton X-100, homogenized in 3–4 one-second bursts with a mechanical homogenizer, and left for 20 min on ice to permeabilize. Next, the cells were pelleted by centrifugation at 120 × *g* for 2 min, resuspended in Isolation solution without triton, and seeded on Poly-L-Lysine-coated (Millipore Sigma) chamber slides (Thermo Scientific). Once settled, the cells were fixed with ice-cold methanol for 1 min and 4% paraformaldehyde for 3 min. For NRVMs, freshly isolated myocytes were plated on nanopatterned coverslips (Nanosurface Cultureware) to promote adult-like myocyte morphology. Forty-eight hours after plating, the myocytes were treated with 2 μM MG132 or equal volume of DMSO. After 24 h, the myocytes were fixed with ice-cold methanol for 1 min, followed by 4% paraformaldehyde for 3 min.

Fixed myocytes were incubated in 0.5% Triton solution for 20 min, followed by 0.1% Triton for 30 min to permeabilize the sarcolemma. Following permeabilization, the cells were incubated in 0.1 M glycine antigen retrieval solution for 30 min, washed three times with PBS, and incubated for 1 h in BSA blocking buffer at room temperature. Next, primary antibodies were added in blocking solution and incubated for 12–14 h at 4 °C. Primary antibodies and dilutions used: BAG3 (Proteintech, 10599-1AP, 1:300), HSPB8 (Proteintech, 15287-1AP, 1:300), HSP70 (Proteintech, 10995-1AP, 1:300), CHIP/STUB1

(Proteintech, 55430-1-AP, 1:300), P62 (Proteintech, 18420-1AP, 1:250), α-actinin (Millipore Sigma, A7811, 1:250), oligomer A11 (Invitrogen, AHB0052, 1:100), Ubiquitin (Cytoskeleton Inc., AUB01, 1:40). After washing four times with PBS, the cells were incubated for one hour with the appropriate secondary antibody conjugated to Alexa-Fluor 488 or Alexa-Fluor 568 (Abcam, 1:1000). Slides were mounted with Vectashield (Vector Laboratories), sealed with coverslips, and imaged at ×63 magnification on a Zeiss LSM 880 super-resolution confocal microscope. Images were acquired under constant laser intensity and photomultiplier gain settings with the Zen Black Software (Zeiss). Quantitative line scan analysis was performed using ImageJ.

**Statistics.** All experiments were performed with more than three independent biological replicates. The results are presented as mean ± SEM and were analyzed using GraphPad Prism 8.0. Datasets containing three or more groups were analyzed by one-way analysis of variance as appropriate. When a significant interaction was identified, Tukey's post-hoc test for multiple comparisons was performed. For comparisons of two groups, data were analyzed by two-tailed Student's *t*-test. A *p*-value < 0.05 was considered as statistically significant.

**Study approval.** The experimental procedures used in the mouse and rat protocols were approved by the Temple University, Hines VA, and Loyola University Chicago Health Science Division IACUCs. Temple University, Hines VA, and Loyola University Chicago Health Science Division are accredited institutions recognized by the Association for Assessment and Accreditation of Laboratory Animal Care (AAALAC). Human samples used for this study were obtained from the tissue biorepositories at the Cleveland Clinic and the Cardiovascular Research Institute at Loyola University Chicago. Informed consent was obtained prior to tissue collection, which was performed with permission from the institutional review boards of Loyola University Chicago and the Cleveland Clinic.

**Reporting summary.** Further information on research design is available in the Nature Research Reporting Summary linked to this article.

## Data availability

The data in support of the findings of this study may be found within the manuscript and in the associated supplementary files. Mass spectrometry-based proteomics raw data are presented in Supplementary Data files 1-3 and were deposited in the MassIVE repository and linked to the ProteomeXchange Consortium with the dataset identifier PXD022414. The raw data and uncropped western blot images pertaining to Figs. 1a–c, e, i–n, 2b–d, f, h, j–l, 4c–g, 5b–d, f, h, j, 6d–m, 7a–c, e–h, j, k, m, and Supplementary Figs. 1a–c, e, 2a–f, 3b–f, 4a–c, e, g, 5a–k, 6b–m are included in the accompanying Source Data File. Data associated with this study will be made available from the corresponding author upon reasonable request.

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

## Acknowledgements

We thank Dr. Jordan Beach from Loyola University Chicago for providing access to the Zeiss LSM 880 in his laboratory, and Peter Caron from Loyola University Chicago for manufacturing custom pieces for our biophysical rigs. This study was supported by the National Institute of Health (HL136737 to J.A.K. and HL91799; HL12309 to A.M.F.) and the American Heart Association (Predoctoral Fellowship 20PRE35170045 to T.G.M.).

## Author contributions

T.G.M. and J.A.K. designed the experiments. T.G.M., V.D.M., P.D., S.D. and E.P. performed the experiments and data analyses. C.S.M. provided human heart tissue samples. M.S.W. provided the BAG3P209L mice. A.M.F. provided tissue from the AAV9/BAG3 and BAG3+/−/BAG3−/− mouse models. J.A.K. provided scientific input from the conception of the project idea through its completion, and substantially contributed to the data presentation. T.G.M. and J.A.K. wrote the manuscript with input from all authors.

## Competing interests

A.M.F. has equity in and is a director of Renovacor, Inc., a biotechnology company developing gene therapy for patients with BAG3 genetic variants. The other authors declare no competing interests.
