## [Peer Review File · Nature Communications]

Reviewers' comments:

Reviewer #1 (Remarks to the Author):

This is a very interesting manuscript on the relation between myofilament BAG3 expression and contractile dysfunction in (dilated) heart failure. An impressive amount of techniques and models have been used, including measurements in human samples, and various animal models. The techniques to assess contractile function in single permeabilized cardiomyocytes are state-of-the-art and the results are well presented.

Major

1. Myofilament BAG3 is part of an autophagy complex localized to the Z-discs which form the boundaries of the sarcomere. The depressed myofilament maximum calcium-activated force (F_{max}) in cardiomyocytes is attributed to sarcomere dysfunction. The sarcomere dysfunction may be attributed to myofibrillar disarray or altered actin-myosin (crossbridge) interaction. The data do not allow to distinguish between these two possibilities; this issue could be addressed.

Minor

2. The temperature at which the force measurements were performed should be stated and examples of the force recordings in cardiomyocytes should be given in the data supplement.
3. It may be a matter of semantics, but: what is the added value of using "predict" in line 46? Also, what do you mean with "impaired" protein turnover in line 8? Increased?

Reviewer #2 (Remarks to the Author):

The authors show that loss of BAG3 function during heart failure impairs quality control mechanisms for myofilament, such as the chaperone-assisted selective autophagy (CASA), thereby inducing sarcomere dysfunction.

General issues:

1. Demonstration of the CASA complex at sarcomere and its dysfunction during heart failure may not be compelling. Repeats of co-immunoprecipitation assays do not necessarily mean that all proteins

are in the same complex, unless they are conducted using a protein complex with a large molecular weight isolated through size fractionation. The result of immunostaining is also insufficient to claim the complex formation at sarcomere. More importantly, the claim that sarcomere proteins are degraded by autophagy is not strong. The presence of p62 in the myofilament fraction alone may not indicate that autophagy is involved.

2. This reviewer was confused by the authors' claim that the level of small HSP, such as HspB8, is decreased when Bag3 is downregulated. The data shown in Fig. 3C and extended Figure 5 do not appear to support this claim. Furthermore, in Fig. 4a, HspB8 in myofilament is actually higher in heart failure. This reviewer was confused by the authors' claim that "Hsp70, HspB8, and CHIP all significantly increased at the myofilament in the control-treated heart failure mice".

3. A recent paper by Ju Chen showed that P209L knock-in mice have no cardiac phenotype up to 16 months old (Fang X et al, AJP Heart 2019). The authors should cite this work and discuss the difference in the phenotype.

Other specific issues:

1. In Fig. 1k, is there a difference in ubiquitin levels in DCM samples grouped by high/low BAG3 expression?
2. In Fig. 3a, is there a difference in ubiquitin levels between the three groups?
3. In Fig. 3a, what is the mechanism by which the level of HSPB8 is decreased in heterozygous and homozygous BAG3 knockout mice? Is the ubiquitin-proteasome pathway involved?
4. In Extended Figs. 2c and 2d, the authors should show the expression level of p62.
5. Figure legend 1d and 1e are mislabeled.

Reviewer #3 (Remarks to the Author):

In their manuscript entitled "BAG3-dependent autophagy maintains sarcomere function in cardiomyocytes", Martin and colleagues study the role of BAG3 in human heart failure using human tissue samples and transgenic mouse models and conclude that BAG3-dependent autophagy is essential for functional maintenance of the cardiac sarcomere. There are two main problems with this manuscript. First, lack of novelty and second lack of mechanistic insights. The authors mainly

confirm previously published data using highly relevant models such as human myocardial LV tissue and mice. Yet, they do not provide any conceptual advance in the understanding of how, mechanistically, BAG3 in association with Hsp70 and HSPB8 is important for myofilament maintenance and how its deregulation contributes to cardiac disease. The authors performed mass spectrometry using myofilament-enriched human LV tissue to identify BAG3-associated proteins. While they confirm the association of BAG3 with its known CASA components HSPB8 and Hsp70, as well as CHIP and p62, they do not report on other specific interactors. One would have expected that such approach would have also allowed the identification of potential substrates of the CASA complex, whose BAG3-mediated clearance is relevant for myofilament maintenance. How can the authors refer to “sarcomere-specific protein quality control” and “binding partners at the sarcomere” without studying any sarcomere-specific interactor? The list of interactors identified by mass spectrometry should have been included in the manuscript and the relevant hits should have been validated functionally.

In addition, some of the conclusions are not experimentally supported. The authors state that “BAG3 overexpression in heart failure restores autophagy flux at the sarcomere”. There is no experimental evidence in support of this claim. In Figure 4, the authors only show that expression levels of p62 and CHIP are similar in myofilament-enriched LV tissue from sham and HF/BAG3 mice, while being enhanced in HF mice. Whether this effect is due to boosting of the autophagic flux is unknown. From the data shown here, it is unclear whether HF reduces the autophagic flux at all. The autophagic flux might not be altered at all in HF mice, and BAG3 overexpression might act by enhancing the delivery of the HSPB8/Hsp70-bound clients to autophagosomes for clearance. The authors try to substantiate their interpretation showing that the levels of ubiquitinated proteins are increased in transgenic mice with cardiomyocyte-specific expression of mutated P209L BAG3. This mutation causes severe dominant childhood muscular dystrophy in humans and has been shown by other groups to affect the ability of BAG3 to release the Hsp70-bound clients for autophagy-mediated clearance. The levels of ubiquitinated proteins are increased in BAG3-P209L expressing cells because of impaired delivery of cargoes to the autophagic system, and not because of defective autophagic flux (Meister-Broekema, *Nat. Commun.* 2018 and Adriaenssens, *BioRxiv.* 2019). Thus, there are no experimental data showing whether/how the autophagic flux is affected in HF and how BAG3 overexpression may influence it.

Other comments

The authors state “Importantly, the functional significance of the CASA complex has never been described in any muscle type.” This is not correct since Arndt et al. reported the implication of BAG3 and its ortholog Starvin in the maintenance of fly muscle by facilitating the clearance of Z disk damaged components, such as filamin. The authors should rephrase their statement. (Arndt, V. et al. *Chaperone-Assisted Selective Autophagy Is Essential for Muscle Maintenance.* *Curr. Biol.* 2010). In this paper, filamin was identified as one of the specific proteins that are targeted by the CASA complex. Thus, it is unclear why the authors did not investigate filamin in their context.

The authors state “To prevent aggregate formation, misfolded proteins are bound by HspB8 and then passed to Hsp70. CASA clients are subsequently ubiquitinated by the E3 ubiquitin ligase CHIP (carboxyl-terminus of Hsp70 interacting protein) and then removed through the actions of the autophagic ubiquitin receptor p62 (SQSTM1), which promotes the association of ubiquitinated proteins with the autophagosome membrane.” This general statement should be rephrased. Both HSPB8 and Hsp70 can directly bind to misfolded proteins and deletion of the HSPB8-binding domain does not prevent BAG3 to target for clearance ubiquitinated substrates bound to Hsp70.

Figure 3: “Proteasome inhibition resulted in a pronounced increase in myofilament protein ubiquitination (Fig. 3d-e). Similar results were obtained for BAG3, Hsp70, and HspB8 (Fig. 3d-e) which all increased significantly at the myofilament.” In figure 3, the authors show immunoblotting of myofilament-enriched lysates from NRVMs treated with MG132 or DMSO control and confirm that proteasome inhibition induces the expression levels of BAG3, Hsp70 and HSPB8, along with total ubiquitinated proteins, as previously published by other groups, using different model systems. These data should be substantiated with confocal microscopy analysis of the subcellular localization of BAG3, Hsp70, HSPB8 and polyubiquitinated proteins.

The authors state "these results implicate CASA as a stress-responsive protein quality control pathway for sarcomere proteins": which are the specific substrates degraded by the CASA complex? Quantitative mass spectrometry may help revealing this, providing new insights into the role of the HSPB8-Hsp70-BAG3 complex in cardiac disease.

Figure 4: “Myofilament levels of P62 also increased in the heart failure mice, indicating impaired autophagic flux”. Measuring only the levels of p62 cannot be used to define the autophagic flux. Although I recognize that measuring autophagic flux in vivo is challenging, other approaches can provide more detailed information. For example, confocal and electron microscopy studies, along with measure of LC3 should substantiate the analysis of p62.

Reviewer #4 (Remarks to the Author):

Both myocardial infarction (MI) as well as other non-ischemic conditions trigger human Heart Failure (HF), leading to dilated cardiomyopathy (DCM). This study examines important mechanisms underlying compromised force-generating capacity in failing cardiomyocytes and protein quality controls (PQC) in DCM; it extends contributions made by previous work (e.g., PMID: 24623017, PMID: 28737513, PMID: 25925243).

Briefly, the authors presented strong evidence documenting the role of myofilament BAG3 expression to rescue myocardial contractile function in the mouse model of HF. Using several “gain-of-function” and “loss-of-function” mouse models, the authors presented evidence that BAG3 deletion and mutation arrested CASA as well as contributed to contractile dysfunction. The study addressed the translational value of the investigation by examining BAG3 in human samples of HF.

Although Nat. Comms. requires flexible format in the first submission, the current version does lack structural clarity. Different sections were mixed together. It was difficult to follow the story with fragmented contents.

Please see the itemized comments below.

Major Concerns:

1. The mouse models of HF were generated via left coronary artery ligation. What are the considerations about these mouse models with respect to their representation on the similarities and differences of ischemic DCM versus that of non-ischemic DCM in humans? Were there any sarcomeric protein mutation(s) found in these clinical samples?

2. What would be the justification on focusing the role of BAG3 in non-ischemic DCM, versus other mouse models of DCM established via genetic mutations on titin (PMID: 28065693), muscle LIM protein (PMID: 28737513), and cardiac troponin (PMID: 17556660)? Should these models be considered in evaluating myofilament BAG3-regulated CASA complex assembly for sarcomere PQC among DCM subtypes?

3. Increased myocardial calcium sensitivities and reduced Fmax were observed in clinical samples of DCM. The mouse models of MI-induced HF and BAG3 genetic perturbation (i.e., BAG3 deletion and BAG3P209L mutation) displayed these parameters somewhat differently. What would be the possible cause for such discrepancy?

4. The autophagy flux was monitored by p62 level and protein ubiquitination. Compromised proteasomal degradation could jointly affect the sarcomeric protein turnover. What are the targets of CASA, as real-time indicators of sarcomeric PQC?

5. What are the relative abundance of complex components, as well as autophagy cargos? These critical information would offer an in-depth picture for the mechanisms underlying the proposed mouse model of DCM and/or human HF.

6. The authors have shown an elevation of ubiquitin level in the mouse model of HF; they attributed this as evidence supporting upregulating sarcomere proteostasis via BAG3-overexpression to restore contractile function. Not sure if level of ubiquitin by itself is sufficient to trigger a compromised autophagy; and what would be the molecular link between recovery of contractile function with restored sarcomere PQC?

Minor Concerns:

1. In Fig. 1e, separating the BAG3 value into quantiles may lose the view of the global association between Fmax and BAG3. Correlation analysis would be a more straightforward way to evaluate the association between them.

2. There seems to be an outlier in Fig. 3a, the level of Hsp70 of homologous BAG3^{-/-} mouse. Additional replicates might be necessary to improve the confidence of this result.

3. The author showed associations between Fmax and BAG3 in all mouse models except the BAG3 deletion model. Is the Fmax level in the BAG3 deleted mouse model of importance to support the role of BAG3 in DCM?

RESPONSE TO REVIEWERS

We thank all reviewers for their thorough evaluation and constructive feedback on our manuscript. In the revised manuscript we have incorporated new experiments, both asked for and others, to directly address reviewer concerns. These added experiments have greatly improved the new manuscript.

Reviewer #1:

1. Myofilament BAG3 is part of an autophagy complex localized to the Z-discs which form the boundaries of the sarcomere. The depressed myofilament maximum calcium-activated force (F_{max}) in cardiomyocytes is attributed to sarcomere dysfunction. The sarcomere dysfunction may be attributed to myofibrillar disarray or altered actin-myosin (crossbridge) interaction. The data do not allow to distinguish between these two possibilities; this issue could be addressed.

Response 1A: The reviewer's concern is appreciated. We acknowledge that while identifying a relationship between impaired/decreased BAG3 function and sarcomere dysfunction, we did not offer a clear mechanism for this, such as may be explained by altered crossbridge interaction or myofibrillar disarray. Previous studies of BAG3 mutations in humans and cell models and of BAG3 KO mice identified impaired protein quality control (specifically failed delivery of substrates to the autophagy system) and severe myofibrillar disarray (Meister-Broekema et al. *Nat Commun*, 2018, Selcen et al. *Ann Neurol*, 2009, Schanzer et al. *Mol Genet Metab*, 2018, Homma et al. *Am J Pathol*, 2006). Without question, such exaggerated structural decline at the sarcomere would lead to impaired functional performance. However, we expect that such drastic structural changes at the sarcomere represent a late stage of disease and are likely a point of no return. In the revised manuscript, we include new functional experiments in 6-week-old BAG3 +/- mice, which at this early time point displayed elevated sarcomere ubiquitination and significantly reduced myofilament force generation (**Figure 2**). Notably, this deficit preceded detectable *in vivo* cardiac dysfunction, which began at 8 weeks (Myers et al. *J Cell Physiol*, 2018). We also did not observe obvious sarcomere structural disarray at 6 weeks in these mice (**Figure 2**). As such, we do not expect the decreased function can be explained solely by sarcomere structural disarray. However, the failure of protein turnover at earlier stages would result in old sarcomere members remaining incorporated in the sarcomere lattice for extended periods of time.

In the revised manuscript, we show ubiquitinated proteins are indeed embedded in the sarcomere using immunofluorescence super-resolution microscopy for ubiquitin in human DCM cardiomyocytes (**Figure 1**). This method revealed that sarcomere proteins incorporated into the complex, particularly in the A-band, are ubiquitinated and together with our western blot data showing increased overall myofilament ubiquitination in these models indicates turnover of the sarcomere proteins is impaired in disease. With enough old/misfolded members remaining in the contractile apparatus, it is likely that contractile dysfunction would occur prior to any noticeable structural disarray.

In the revised manuscript we also performed a novel mass spectrometry experiment in which we enriched for ubiquitinated sarcomeric peptides in human heart failure and the mouse heart failure model to determine which proteins have impaired turnover in disease (**Figures 1 & 7**). We found many sarcomere proteins that had increased ubiquitination in heart failure, including key proteins involved in contraction and sarcomere structural integrity. When we assessed the myofilament ubiquitinome in the heart failure mice treated with BAG3 gene therapy, we identified four sarcomere clients for BAG3-mediated degradation that displayed reduced ubiquitination after treatment, suggesting rescued turnover of these proteins restored force-generating capacity.

2. *The temperature at which the force measurements were performed should be stated and examples of the force recordings in cardiomyocytes should be given in the data supplement.*

Response 1B: Thank you. The force measurements were performed at room temperature (22-23 °C). We have included an example force recording in the revised submission (**Supplemental Figure 4**).

3. *It may be a matter of semantics, but: what is the added value of using “predict” in line 46? Also, what do you mean with “impaired” protein turnover in line 8? Increased?*

Response 1C: We acknowledge that our wording could have been made clearer in the initial submission. These concerns have been addressed in the revised manuscript. By impaired protein turnover we meant decreased turnover.

Reviewer #2:

1. Demonstration of the CASA complex at sarcomere and its dysfunction during heart failure may not be compelling. Repeats of co-immunoprecipitation assays do not necessarily mean that all proteins are in the same complex, unless they are conducted using a protein complex with a large molecular weight isolated through size fractionation. The result of immunostaining is also insufficient to claim the complex formation at sarcomere. More importantly, the claim that sarcomere proteins are degraded by autophagy is not strong. The presence of p62 in the myofilament fraction alone may not indicate that autophagy is involved.

Response 2A: We thank the reviewer for their thoughtful critique. In the revised manuscript we not only show that the CASA complex is at the sarcomere, but using a sophisticated mass spectrometry approach to enrich for ubiquitinated peptides (Udeshi et al. *Nat Protoc*, 2013) we identified four sarcomeric protein clients of BAG3-mediated protein turnover (**Figure 7**). These findings advance our understanding of BAG3's role in the heart and help to explain the sarcomeric functional restoration with BAG3 gene therapy.

Regarding the assembly of the CASA complex at the cardiac sarcomere, we acknowledge that their co-localization and co-immunoprecipitation with BAG3 alone are not enough to validate assembly of the entire complex at one time. However, given the previous finding of the CASA complex at the Z-disc in skeletal muscle, it is highly likely that their co-localization and co-IP with BAG3 indicates the complex is present at the cardiac sarcomere.

With respect to the immuno images not fully validating BAG3/CASA is at the sarcomere, we agree with the reviewer that immuno alone is insufficient to confirm sarcomeric localization of the complex. However, the immuno taken together with western blot data from myofilament enriched protein lysates and myofilament BAG3 interactome assessment by mass spectrometry (which are included in the revised manuscript) strongly indicate the complex is present at the sarcomere (**Figure 3**).

The reviewer makes an important point that the protein degradation pathway involved is not necessarily autophagy according to the data provided in our manuscript. We acknowledge that in our initial submission the language used regarding autophagy was too strong. While it is likely to be autophagy given the many previous studies linking BAG3 and the other CASA members to autophagy, we did not explicitly show the involvement of autophagy in our study. What is clear in our study is that there is a protein quality control problem at the sarcomere in heart failure and with BAG3 deficiency. Moreover, we have shown that increasing BAG3 levels by adenovirus gene therapy was sufficient to restore sarcomere protein turnover and restored function (**Figure 6**). To add mechanistic insight, in the revised manuscript we performed a ubiquitinated peptide enrichment from the myofilament fraction to determine which specific clients have impaired turnover in heart failure. By this method we identified several interesting candidates, four of which displayed increased turnover with BAG3 gene therapy (**Figure 7**). In the new manuscript, we have softened the terminology around autophagy while clearly delineating the requirement of BAG3 for sarcomere protein turnover and function.

2. This reviewer was confused by the authors' claim that the level of small HSP, such as HspB8, is decreased when Bag3 is downregulated. The data shown in Fig. 3C and extended Figure 5 do not appear to support this claim. Furthermore, in Fig. 4a, HspB8 in myofilament is actually higher in heart failure. This reviewer was confused by the authors' claim that "Hsp70, HspB8, and CHIP all significantly increased at the myofilament in the control-treated heart failure mice".

Response 2B: We appreciate the reviewer’s concern regarding HspB8 expression with downregulation of BAG3. Support for our claim that HspB8 decreases with decreased myofilament BAG3 can be found in the BAG3 +/- and KO mouse data which showed a prominent decrease in myofilament HspB8 in both the BAG3 +/- mice and KO mice (**Figure 5**). Additionally, in the human samples used for this study, when we separated into highest and lowest BAG3 expressors, we found again that HspB8 was significantly decreased in the “low BAG3” group (**Figure 5**). While this comparison is purely correlative, when taken together with the mouse KO data, it is quite clear that BAG3 is required for localization of HspB8 at the myofilament, which we show is relevant in the human heart as well.

3. A recent paper by Ju Chen showed that P209L knock-in mice have no cardiac phenotype up to 16 months old (Fang X et al, *AJP Heart* 2019). The authors should cite this work and discuss the difference in the phenotype.

Response 2C: The reviewer makes an important suggestion regarding inclusion of the P209L (P215L in mouse) paper from Ju Chen’s group. In the transferred format of the initial submission, ample space was not available to discuss many important earlier studies, such as the Fang et al. paper. In the revised manuscript, which is now written in the Nature Comms style, we have included a discussion of the P215L mice and our human P209L BAG3 transgenic mice, and explain why differences may be expected in the cardiac phenotype of these two models (**pages 19-20**). The primary reason to expect differences was stated in the Fang et al. paper, where the authors posit that the pathogenicity of the mutation may be specific to the human isoform. Importantly, in our model which expresses the human mutant BAG3 transgene, the mice display a restrictive cardiomyopathy phenotype which mimics that found in human patients with the P209L mutation (Quintana et al. *Am J Pathol*, 2016).

Other specific issues:

1. In Fig. 1k, is there a difference in ubiquitin levels in DCM samples grouped by high/low BAG3 expression?

Response 2D: We thank the reviewer for the suggestion to assess myofilament ubiquitin by high/low BAG3 in the human samples. When separating ubiquitin levels in the DCM samples by high/low BAG3 expression, we did not find a significant difference. However, this is perhaps not unexpected as the patient samples used come from varying stages of heart failure (NYHA class II to class IV). In response to sarcomere stress, BAG3/CASA are targeted to the myofilament fraction as we have shown in NRVMs. Our data in the mouse model support that in the early stages of heart failure this targeting of BAG3/CASA to the myofilament fraction in response to elevated stress/proteotoxicity is conserved. However, we show the CASA members aggregate at the sarcomere in the early progression to heart failure due to inadequate clearance (**Figure 7**). We hypothesize that BAG3 decreases in end-stage heart failure due to an ultimate failure of sarcomeric protein quality control mechanism to upregulate BAG3 in response to the chronic level of proteotoxic stress therein. Given such apparently heart failure stage-specific differences in CASA targeting to the sarcomere, we do not expect that

sarcomeric expression of these members directly correlates with myofilament protein ubiquitination unless samples are taken solely from the end-stage of the disease.

2. *In Fig. 3a, is there a difference in ubiquitin levels between the three groups?*

Response 2E: The WT, +/-, and KO mice were included in this study as a tool to determine the requirement of BAG3 for HspB8, CHIP, and Hsp70 localization to the sarcomere. As such, these mice were only 6 weeks of age at time of euthanasia, which is two weeks prior to any detectable cardiac dysfunction (Myers et al. *J Cell Physiol*, 2018). However, when we assessed myofilament ubiquitin in the mice we did find a mild but significant increase in the BAG3^{+/-} mice compared with WT (**Figure 2**). Interestingly, we could also identify decreased sarcomere force-generating capacity at this timepoint in the BAG3^{+/-} mice (included in the revised manuscript). Changes in ubiquitin in the KO mice were not significant as a result of low power due to smaller sample size (n = 3).

3. *In Fig. 3a, what is the mechanism by which the level of HSPB8 is decreased in heterozygous and homozygous BAG3 knockout mice? Is the ubiquitin-proteasome pathway involved?*

Response 2F: The mechanism by which HspB8 decreases in the BAG3 KO mice is not clear in our present study. However, earlier work suggests BAG3 is required to maintain stability of small HSPs, so this offers a possible explanation (Fang et al., *JCI*, 2018).

4. *In Extended Figs. 2c and 2d, the authors should show the expression level of p62.*

Response 2G: We appreciate the reviewer's request for P62 in the 6-week-old WT, +/-, and KO mice. For the same reasons mentioned in response 2E, these analyses were not performed in the initial manuscript. In the revised manuscript, we have moved away from our focus on autophagy-specific assessment and thus did not find assessment of P62 in this model fit well into our new story.

5. *Figure legend 1d and 1e are mislabeled.*

Response 2H: Thank you for pointing out. This has been addressed.

Reviewer #3:

1. *There are two main problems with this manuscript. First, lack of novelty and second lack of mechanistic insights. The authors mainly confirm previously published data using highly relevant models such as human myocardial LV tissue and mice. Yet, they do not provide any conceptual advance in the understanding of how, mechanistically, BAG3 in association with Hsp70 and HSPB8 is important for myofilament maintenance and how its deregulation contributes to cardiac disease.*

Response 3A: We understand the reviewer's concern regarding lack of mechanistic insight in the initial manuscript. This has been addressed in the revised manuscript. In our initial work, we showed myofilament protein ubiquitination increased in heart failure (human DCM and mouse model) but was restored to baseline levels by BAG3 gene therapy. We attributed the functional decline in heart failure to impaired turnover of sarcomeric proteins, which BAG3 appeared to rectify. In the revised manuscript we further show the importance of BAG3 for sarcomere function using 6-week-old BAG3^{+/-} mice (**Figure 2**). These mice, which we previously showed develop cardiac dysfunction by 8 weeks (Myers *et al. J Cell Physiol*, 2018), had increased myofilament protein ubiquitination and significantly reduced myofilament force generating capacity. These data further suggest a link between impaired sarcomere protein turnover in the BAG3 deficient mice that causes functional decline. However, they do not identify which proteins are clients for BAG3-mediated turnover.

To determine which sarcomere proteins are clients of BAG3/CASA, we used a novel quantitative mass spectrometry approach in which ubiquitinated peptides were enriched for using the ubiquitin remnant motif antibody. We first characterized the myofilament ubiquitinome in human DCM and identified numerous proteins with elevated ubiquitination, including several directly involved in sarcomere contraction (**Figure 1**). Importantly, these ubiquitinated sarcomere proteins are not components of protein aggregates as have been previously observed with decreased/dysfunctional BAG3, but are physically embedded into the sarcomere itself, which we confirm by super resolution microscopy in the revised manuscript (**Figure 1**). This is the first report of which specific sarcomere proteins have impaired turnover in DCM and offers a compelling explanation for the reduced force-generation that is commonly found in cardiomyocytes from DCM patients. Importantly, when we repeated this procedure in the mouse heart failure model many of the ubiquitinated candidates were shared with the human samples (**Figure 7**). However, in the heart failure mice that received AAV9/BAG3 four of these proteins displayed significantly reduced ubiquitination, indicating restored turnover (**Figure 7**). One of the BAG3 clients identified was filamin C, as had previously been identified as a CASA client skeletal muscle. However, the other three have never previously been reported as BAG3/CASA clients and are directly involved in either sarcomere structural integrity or the cross-bridge cycle. These findings provide important mechanistic insight into BAG3's role at the sarcomere and represent a conceptual advance that was lacking in the initial submission.

2. *The authors performed mass spectrometry using myofilament-enriched human LV tissue to identify BAG3-associated proteins. While they confirm the association of BAG3 with its known CASA components HSPB8 and Hsp70, as well as CHIP and p62, they do not report on other specific interactors. One would have expected that such approach would have also allowed the identification of potential substrates of the CASA complex, whose BAG3-mediated clearance is relevant for myofilament maintenance. How can the authors refer to "sarcomere-specific protein quality control" and "binding partners at the sarcomere" without studying any sarcomere-specific interactor? The list of interactors identified by mass spectrometry should have been included in the manuscript and the relevant hits should have been validated functionally.*

Response 3B: Thank you for pointing out. The full list of BAG3-associated proteins identified by mass spectrometry was not included in the initial manuscript due to the method used. We initially used an in-gel digestion protocol where only two distinct bands were cut out and run on LC-MS/MS: one at 70-80 kDa and the other at 25-30 kDa. This method, while helpful in identifying the interactors in the two most prominent bands, does not allow for accurate assessment of the entire interactome. This has been rectified in the revised manuscript. We now use an in-solution digestion technique of all proteins that were co-immunoprecipitated in the myofilament fraction with BAG3. By this method we again identified HSPB8 and HSP70 among the top hits and identified many potential clients for BAG3/CASA-mediated degradation, including the clients we identified by ubiquitin-enrichment mass spectrometry in the human DCM and mouse HF/BAG3 model. The full table of BAG3-associated proteins is provided in the data supplement of the revised manuscript (**Supplemental Table 1**).

3. In addition, some of the conclusions are not experimentally supported. The authors state that "BAG3 overexpression in heart failure restores autophagy flux at the sarcomere". There is no experimental evidence in support of this claim.

Response 3C: We acknowledge the reviewer's point that our claims of BAG3 gene therapy restoring autophagy flux in heart failure were not fully borne out in the data. We also agree with the reviewer that "autophagy flux" was not directly measured, as it can be very challenging to characterize *in vivo*. In the revised manuscript, we have moved away from this terminology. The characterization of the ubiquitinated proteins in heart failure by ubiquitin-enrichment LC-MS/MS now provide experimental evidence that BAG3 overexpression heart failure restores sarcomere protein turnover and identifies specific sarcomere substrates for BAG3/CASA (**Figure 7**).

4. In Figure 4, the authors only show that expression levels of p62 and CHIP are similar in myofilament-enriched LV tissue from sham and HF/BAG3 mice, while being enhanced in HF mice. Whether this effect is due to boosting of the autophagic flux is unknown. From the data shown here, it is unclear whether HF reduces the autophagic flux at all. The autophagic flux might not be altered at all in HF mice, and BAG3 overexpression might act by enhancing the delivery of the HSPB8/Hsp70-bound clients to autophagosomes for clearance. The authors try to substantiate their interpretation showing that the levels of ubiquitinated proteins are increased in transgenic mice with cardiomyocyte-specific expression of mutated P209L BAG3. This mutation causes severe dominant childhood muscular dystrophy in humans and has been shown by other groups to affect the ability of BAG3 to release the Hsp70-bound clients for autophagy-mediated clearance. The levels of ubiquitinated proteins are increased in BAG3-P209L expressing cells because of impaired delivery of cargoes to the autophagic system, and not because of defective autophagic flux (Meister-Broekema, Nat. Commun. 2018 and Adriaenssens, BioRxiv. 2019). Thus, there are no experimental data showing whether/how the autophagic flux is affected in HF and how BAG3 overexpression may influence it.

Response 3D: We appreciate the reviewer's concern. In the initial manuscript too much of the terminology focused around "flux," which as the reviewer indicates was not sufficiently measured. Whether the BAG3 overexpression restored autophagic flux at the sarcomere, enhanced the delivery of protein clients to the autophagosomes, or both is not clear. However, the ubiquitin enrichment mass spectrometry results in the revised manuscript (discussed above) unequivocally show that BAG3 gene therapy restores turnover of several sarcomere clients and offers an explanation for how function was recovered. As the reviewer mentions, the P209L mutation impairs delivery of cargo to the autophagy system and not flux as we had mistakenly referred. We show with this model that ubiquitinated sarcomere proteins are elevated, indicating impaired clearance as previously

characterized mechanistically. Importantly, this impaired clearance was again linked to decreased myofilament contractile function in our data (**Supplemental Figure 2**). While we do not in this manuscript distinguish between autophagic flux and delivery of ubiquitinated cargo to the autophagosomes, we hope the reviewer will acknowledge that BAG3 gene therapy restored sarcomere proteostasis through enhancing turnover of specific sarcomere clients, which provides a compelling explanation for the functional recovery (**Figures 6 & 7**). We have removed references to autophagic flux from the revised manuscript as this was never fully characterized.

5. The authors state *“Importantly, the functional significance of the CASA complex has never been described in any muscle type.”* This is not correct since Arndt et al. reported the implication of BAG3 and its ortholog Starvin in the maintenance of fly muscle by facilitating the clearance of Z disk damaged components, such as filamin. The authors should rephrase their statement. (Arndt, V. et al. Chaperone-Assisted Selective Autophagy Is Essential for Muscle Maintenance. Curr. Biol. 2010). In this paper, filamin was identified as one of the specific proteins that are targeted by the CASA complex. Thus, it is unclear why the authors did not investigate filamin in their context.

Response 3E: Thank you. We are aware of the Arndt et al. paper and should have been clearer in our statement, which was intended to delineate that the myofilament functional significance (not structural significance) had not previously been assessed. We have revised the text in the new manuscript. We also investigated filamin C in the revised manuscript by the ubiquitin-enrichment mass spectrometry where we show that BAG3 gene therapy decreases ubiquitinated filamin-C levels, indicating enhanced clearance. However, we also identified three other novel candidates for BAG3-mediated protein degradation that were not previously identified in the Arndt paper or any other study to this point (**Figure 7**).

6. The authors state *“To prevent aggregate formation, misfolded proteins are bound by HspB8 and then passed to Hsp70. CASA clients are subsequently ubiquitinated by the E3 ubiquitin ligase CHIP (carboxyl-terminus of Hsp70 interacting protein) and then removed through the actions of the autophagic ubiquitin receptor p62 (SQSTM1), which promotes the association of ubiquitinated proteins with the autophagosome membrane.”* This general statement should be rephrased. Both HSPB8 and Hsp70 can directly bind to misfolded proteins and deletion of the HSPB8-binding domain does not prevent BAG3 to target for clearance ubiquitinated substrates bound to Hsp70.

Response 3F: Thank you. This issue has been addressed in the revised manuscript.

7. Figure 3: *“Proteasome inhibition resulted in a pronounced increase in myofilament protein ubiquitination (Fig. 3d-e). Similar results were obtained for BAG3, Hsp70, and HspB8 (Fig. 3d-e) which all increased significantly at the myofilament.”* In figure 3, the authors show immunoblotting of myofilament-enriched lysates from NRVMs treated with MG132 or DMSO control and confirm that proteasome inhibition induces the expression levels of BAG3, Hsp70 and HSPB8, along with total ubiquitinated proteins, as previously published by other groups, using different model systems. These data should be substantiated with confocal microscopy analysis of the subcellular localization of BAG3, Hsp70, HSPB8 and polyubiquitinated proteins.

Response 3G: We appreciate the reviewer’s concern that these findings be further confirmed by another method. We have repeated the experiment using confocal microscopy as the reviewer suggests and show that BAG3 increases specifically at the Z-disc after proteasome inhibition (**Figure 4**). Together with the myofilament enriched lysates, this data supports an upregulation of the CASA machinery at the sarcomere in response to proteotoxicity.

8. The authors state "these results implicate CASA as a stress-responsive protein quality control pathway for sarcomere proteins": which are the specific substrates degraded by the CASA complex? Quantitative mass spectrometry may help revealing this, providing new insights into the role of the HSPB8-Hsp70-BAG3 complex in cardiac disease.

Response 3H: Thank you for this excellent suggestion. This has been addressed with the ubiquitin-enrichment LC-MS/MS as discussed above.

9. Figure 4: "Myofilament levels of P62 also increased in the heart failure mice, indicating impaired autophagic flux". Measuring only the levels of p62 cannot be used to define the autophagic flux. Although I recognize that measuring autophagic flux in vivo is challenging, other approaches can provide more detailed information. For example, confocal and electron microscopy studies, along with measure of LC3 should substantiate the analysis of p62.

Response 3I: We acknowledge the reviewer's concern regarding statements of autophagic flux. As mentioned above, we have moved away from this specific terminology in the present manuscript.

Reviewer #4:

1. *Although Nat. Comms. requires flexible format in the first submission, the current version does lack structural clarity. Different sections were mixed together. It was difficult to follow the story with fragmented contents.*

Response 4A: We apologize for the lack of cohesiveness in the initial manuscript. Our format for the first manuscript was transferred from another Nature journal, which limited content to 2000 words. The revised manuscript is now in the proper Nature Comms format, which allows for more structural clarity.

2. *The mouse models of HF were generated via left coronary artery ligation. What are the considerations about these mouse models with respect to their representation on the similarities and differences of ischemic DCM versus that of non-ischemic DCM in humans? Were there any sarcomeric protein mutation(s) found in these clinical samples?*

Response 4B: This is an important question. Little is known regarding the specific representation of the MI mouse model and various etiologies of DCM. However, for our purposes, which were to study the protein quality control impairment that occurs in DCM and determine if BAG3 gene therapy could restore function, the mouse model nicely matches the human non-ischemic DCM. This is supported by new data in the revised manuscript from a novel mass spectrometry approach with immunoaffinity purification in which we assessed ubiquitinated peptides from the human DCM samples and the mouse heart failure samples. By this approach we characterized the myofilament ubiquitinome in both of these models and found many of the same candidates had impaired turnover (i.e. elevated ubiquitination) compared to non-failing/sham (**Figures 1 & 7**).

As for the protein mutations in the clinical samples, there were none reported among the samples used. However, this is an evolving area and many patients still are not tested for genetic risk factors, though work from the Seidman lab suggests that many patients with DCM have titin mutations. We cannot rule this possibility out in these samples.

2. *What would be the justification on focusing the role of BAG3 in non-ischemic DCM, versus other mouse models of DCM established via genetic mutations on titin (PMID: 28065693), muscle LIM protein (PMID: 28737513), and cardiac troponin (PMID: 17556660)? Should these models be considered in evaluating myofilament BAG3-regulated CASA complex assembly for sarcomere PQC among DCM subtypes?*

Response 4C: We appreciate the reviewer's concern regarding different models of DCM. Our hypothesis was that BAG3 gene therapy could restore myofilament force generating capacity in heart failure, and in the MI mouse model previous evidence showed BAG3 gene therapy restored *in vivo* function. Therefore, this model seemed the most suitable place to start to answer our question.

Mouse models of DCM established by genetic mutations create complex questions. For one, these mutations may have their own negative impact on contractile function that is separate from a protein quality control question. On the other hand, mutations in these proteins may affect their own protein turnover mechanisms in ways unique to each mutation (Glazier et al. JCI Insight, 2018, Helms et al. JCI Insight, 2020). In order to avoid such complexity, which would make interpreting our results more challenging, we chose the MI model.

3. *Increased myocardial calcium sensitivities and reduced Fmax were observed in clinical samples of*

DCM. The mouse models of MI-induced HF and BAG3 genetic perturbation (i.e., BAG3 deletion and BAG3P209L mutation) displayed these parameters somewhat differently. What would be the possible cause for such discrepancy?

Response 4C: The reviewer asks an important question. In end-stage heart failure, increased calcium sensitivity and reduced Fmax are common. However, we chose relatively early time points in our animal models to study, at which point dysfunction would be less severe. For example, the P209L mutation mice only just develop *in vivo* cardiac dysfunction at the 8-month time point, which we used for myofilament function experiments. In our mouse MI model we chose a time-point 8 weeks following infarction, which may yet represent an earlier stage in the progression to heart failure. The early time points were chosen to avoid end-stage heart failure, where complete sarcomere structural disarray has occurred, as we were concerned sarcomere function may not be recoverable at such a late stage. Thus, we expect the calcium sensitivity would increase further in the various mouse models if the disease progressed. Another possibility that has been debated in the field is that calcium sensitivity increases in human HF samples because of decreased PKA phosphorylation of troponin I that arises from beta-blocker therapy – an intervention absent in mouse models.

Future work will focus on the optimal timing of BAG3 gene therapy as an intervention and determine how much disarray is recoverable.

4. The autophagy flux was monitored by p62 level and protein ubiquitination. Compromised proteasomal degradation could jointly affect the sarcomeric protein turnover. What are the targets of CASA, as real-time indicators of sarcomeric PQC? What are the relative abundance of complex components, as well as autophagy cargos? These critical information would offer an in-depth picture for the mechanisms underlying the proposed mouse model of DCM and/or human HF.

Response 4D: We appreciate the reviewer's interest in the sarcomeric clients for BAG3-mediated degradation. Using a sophisticated mass spectrometry approach, in the revised manuscript we identified four different clients with roles in sarcomere structural integrity and the cross-bridge cycle as discussed above (**Figure 7, Responses 1A & 3A**). These findings provide a mechanistic explanation for the sarcomere functional recovery with BAG3 gene therapy.

5. The authors have shown an elevation of ubiquitin level in the mouse model of HF; they attributed this as evidence supporting upregulating sarcomere proteostasis via BAG3-overexpression to restore contractile function. Not sure if level of ubiquitin by itself is sufficient to trigger a compromised autophagy; and what would be the molecular link between recovery of contractile function with restored sarcomere PQC?

Response 4E: Thank you. We expect that the increased ubiquitination is not necessarily a trigger for impaired autophagy, but rather evidence that the turnover of sarcomere proteins is either stalled or is occurring at an inadequate rate. The molecular link between impaired turnover of sarcomere proteins and decreased contractile force capacity is discussed in **Response 1A** above.

6. In Fig. 1e, separating the BAG3 value into quantiles may lose the view of the global association between Fmax and BAG3. Correlation analysis would be a more straightforward way to evaluate the association between them.

Response 4F: We understand and agree. However, while the correlation analysis does show a significant relationship, it also appears less convincing because of the variability in the 3rd quartile of data. Its thus not likely to be a linear relationship, since a modest loss of BAG3 may not be pathologic.

7. *There seems to be an outlier in Fig. 3a, the level of Hsp70 of homologous BAG3^{-/-} mouse. Additional replicates might be necessary to improve the confidence of this result.*

Response 4G: We agree that one of the Hsp70 points appears to be an outlier. However, even when removed, Hsp70 displays only a modest decrease (~25%) compared to WT. While this may be significant if additional replicates were performed, we note that the majority of Hsp70 localized to the sarcomere does not require BAG3's presence. This is expected given Hsp70's numerous roles in protein complex assembly and turnover at the sarcomere (Willis et al. *Cardiovasc Res*, 2009).

8. *The author showed associations between Fmax and BAG3 in all mouse models except the BAG3 deletion model. Is the Fmax level in the BAG3 deleted mouse model of importance to support the role of BAG3 in DCM?*

Response 4H: We thank the reviewer for making this suggestion. In the revised manuscript, we include myofilament functional data for the WT and BAG3^{+/-} mice, in which we found significantly decreased Fmax by 6 weeks of age. Notably, this functional decline again occurred with a concurrent increase in sarcomere protein ubiquitination (**Figure 2**).

REVIEWER COMMENTS

Reviewer #1 (Remarks to the Author):

All my points have been addressed satisfactorily.

Reviewer #2 (Remarks to the Author):

The revised paper is improved but the data may not fully support the conclusions in this work.

In Figure 1HI and 7I-O, the authors should evaluate the total protein level and the amount of each ubiquitinated peptide should be normalized by the total protein level.

What is the mechanism by which myofilament proteins are ubiquitinated in DCM hearts? Since the authors propose that downregulation of BAG3 inhibits the assembly of the CASA, it is unlikely that it is mediated by CHIP. In addition, what is the evidence to support that ubiquitinated proteins are dysfunctional?

In Figure 6, why addition of BAG3 rather decreases ubiquitination of the myofilament despite the fact that it promote CASA assembly at the myofilament in mice?

Why is the assembly of the CASA complex in the myofilament fraction opposite (one disappears and the other stays) between human DCM and mouse heart failure?

According to Figure 7A, the effect of AAV-BAG3 upon the level of total BAG3 is negligible. Why can it induce a significant effect?

In Figure 5J, one cannot say anything about the level of CHIP with $p=0.21$.

Reviewer #3 (Remarks to the Author):

The manuscript has substantially improved and all the concerns have been addressed either experimentally or through textual revision.

Reviewer #4 (Remarks to the Author):

Remarkable effort has been made to improve this manuscript. In the revised version, authors largely extended the scope of original study, adding adequate new data obtained via proteomics, functional assessment, and molecular imaging approaches. The affinity-based proteomics approach was innovatively employed to capture sarcomeric protein targets of ubiquitination in failing hearts of mice and humans. A label-free proteomic method was used to differentiate protein ubiquitination levels in HF mice with and without AAV9/BAG3 treatment, identifying 3 sarcomeric proteins as novel BAG3/CASA clients in cardiac tissue. Authors also modified their original in-gel digestion protocol to better characterize the BAG3 interactome in myofilament. The revised experiments identified 40+ BAG3 associating partners as potential clients of BAG3/CASA-mediated protein degradation. By adding these new results, the revised manuscript presents strong and coherent evidence supporting the proposed mechanism. This new version of the manuscript was largely restructured and rewritten; its current format is in compliance with the journal's guidelines for original research articles. However, there are still several minor issues that you may consider to rephrase during the proof-reading.

Minor Comments:

P3, Line 81-82: "An elevation of dysfunctional proteins incorporated into the sarcomere in heart failure...". These dysfunctional proteins were denatured by mechanical stress; they are existing components of sarcomere. Could "An arrested removal of dysfunctional proteins led to their aggregations in the sarcomere in heart failure" better describe the mechanism of decreased F_{max} ?

P5, Line 118: "where lowest BAG3 expression predicted weakest force generation". Both BAG3 expression level and force generation capacity were measured for human DCM samples. Should these two parameters be described as "correlated" instead of "predicted"?

The manuscript could benefit from another round of editing to elevate its readability. We noticed a few cases, such as adding “that” after “indicates” on P5 Line 172, moving “by western blot” after “were found” on P5 Line 173, and adding “that” after “suggests” on P5 Line 175.

We understand the author’s argument on not calculating correlation between Fmax and BAG3 level for the reason that the relationship may not be linear and a modest loss of BAG3 may not be pathologically important. However, we believe that a scatter plot can address the problems and better demonstrate the trend.

RESPONSE TO REVIEWERS

We thank the reviewers again for their critical analysis of our newly revised and formatted manuscript. While reviewers 1 and 3 acknowledged that the new manuscript fully addressed their concerns, reviewers 2 and 4 made additional requests toward the goal of strengthening the manuscript and its conclusions. In this newly revised submission, we address those concerns through additions to the text and elaboration on our findings. Additionally, we have reanalyzed portions of our data and performed new functional experiments at the request of reviewer 2, which we think strengthen the overall conclusions of the manuscript that dysfunctional/misfolded sarcomere proteins remaining in the contractile apparatus in heart failure and depress force-generating capacity.

Reviewer #1:

All my points have been addressed satisfactorily.

Reviewer #2:

1. In Figure 1HI and 7I-O, the authors should evaluate the total protein level and the amount of each ubiquitinated peptide should be normalized by the total protein level.

Response 2A. We appreciate the reviewer's point that ubiquitinated peptide level be normalized by total peptide level. However, since these are sarcomere proteins that we are considering, which we show are ubiquitinated and remain integrated into the sarcomere, they are expected to maintain a strict stoichiometry and required of sarcomere members (Thompson and Metzger, *Anat Rec (Hoboken)*, 2015). Therefore, normalizing them to their own total peptide amounts should not differ from normalizing to overall total peptides (as we have done). To confirm this, in the revised submission we assessed the total protein level for each ubiquitinated protein, and found they did not change between non-failing and DCM in the humans, or between the sham, HF, and HF/BAG3 models in the mice (shown in new **Supplemental Figures 2 & 6**). The one exception, however, was desmin in the mouse HF model, which had increased peptides compared with sham. This finding is not surprising as desmin is an intermediate filament protein that does not adhere to the strict stoichiometric requirements of the sarcomere proteins and has previously been shown to aggregate in heart failure (Singh and Robbins, *Circulation Research*, 2020). Desmin was also not concluded to be a definitive client of CASA in our data. Given this data, we have maintained the current normalization to overall total peptides for the proteomics data, but have added two supplemental figures to show the protein levels are unchanged.

We thank the reviewer for this comment, as it also allowed us to strengthen our data showing that the dysfunction comes from misfolded/ubiquitinated proteins still embedded in the sarcomere lattice, and therefore protein levels do not change.

2. What is the mechanism by which myofilament proteins are ubiquitinated in DCM hearts? Since the authors propose that downregulation of BAG3 inhibits the assembly of the CASA, it is unlikely that it is mediated by CHIP.

Response 2B. We appreciate this important question from the reviewer, as we have wondered this as well. We agree that CHIP is unlikely the primary mediator of myofilament protein ubiquitination. While our data clearly suggests BAG3 is at least partially required for CHIP localization to the sarcomere, it is probable the other E3 ligases are mediating some of the myofilament protein ubiquitination observed. However, this should not come as a surprise as over a dozen E3 ligases have been identified to mediate ubiquitination of sarcomere proteins (Martin and Kirk, *J Mol Cell Cardiol*, 2020). Moreover, HSP70 clients themselves have also been identified in recent years to be ubiquitinated by E3 ligases other than CHIP, including Mdm2, Sis1, and UBR1 (Boysen *et al.*, *Mol. Cell*, 2019; Ho *et al.*, *Nat. Comms.*, 2019; Nillegoda *et al.*, *Mol. Biol. Cell*, 2010). Thus, ubiquitination of HSP70 clients may occur even in the absence of CHIP through these other ubiquitin ligases. While the canonical CASA pathway involves the complex binding to misfolded proteins that are *then* ubiquitinated by CHIP, it has been previously shown in neurons that BAG3 also mediates autophagic degradation of proteins that have *already* been ubiquitinated but not removed (Minoia *et al.*, *Autophagy*, 2014). Therefore, it is likely that CHIP is involved in the clearance of some CASA/BAG3 clients, but not all.

In the revised manuscript, to directly address the reviewer's question, we performed mass spectrometry analysis of the HSP70 myofilament interactome and identified 2 additional ubiquitin ligases, UBA52 and MIB2 (**Supplementary Data 2**), and also identified 7 other E3 ligases in the BAG3 myofilament interactome (**Supplementary Data 1**). All this to say that the specific regulation of HSP70 client ubiquitination and the timing of the CASA complex clearance (whether it binds only to un-ubiquitinated misfolded proteins and then ubiquitinates them, or also clears pre-ubiquitinated misfolded proteins as previously identified) are not clear. However, what is clear from our data is that this process is certainly more complicated than just CHIP mediating all ubiquitination of sarcomere proteins. Regardless, our data indicate that in the absence of BAG3 or when BAG3's functions are compromised (HF, P209L mutation), ubiquitinated sarcomere proteins increase significantly. However, when BAG3 was increased in HF, the sarcomere ubiquitination decreased, indicating a restoration of proteostasis. These data solidify BAG3 as an essential protein for sarcomere protein turnover.

In the revised manuscript we have elaborated on our earlier finding and new data and incorporated a fuller discussion of existing literature throughout to better interpret the findings related to increased ubiquitinated with dysfunctional/decreased BAG3.

3. In addition, what is the evidence to support that ubiquitinated proteins are dysfunctional?

Response 2C. We acknowledge the reviewer's concern that our manuscript provides no direct evidence that the ubiquitinated proteins are dysfunctional. However, we apologize for not being clear, our hypothesis is that misfolded proteins that remain embedded in the sarcomere are dysfunctional, for which elevated ubiquitination serves as an appropriate readout (Houck *et al.*, *Methods Mol. Biol.*, 2012). We do not necessarily claim there is a functional impact of the ubiquitin modification itself, although we have not ruled out this possibility.

However, to provide evidence that sarcomere protein misfolding alone is sufficient to cause a depression in the force-generating capacity of the myofilament, we performed a new functional experiment in the revised manuscript. Here, we took *skinned* LV cardiomyocytes from the non-failing human samples and incubated them with mild heat shock at 43 °C, a common approach to cause protein denaturation/misfolding. Importantly, this was done in skinned myocytes, so we are only looking at the myofilament lattice here, and not a cellular response to heat shock. We then performed

force-calcium functional experiments on these myocytes and controls from the same samples. As expected, we found that heat shock caused significant reduction in F_{max} (**Supplemental Figure 1**). Importantly, we incubated the myocytes with protease inhibitors, so these findings are not due to protein degradation. Also, since these were skinned myocytes, there was no induction of protein ubiquitination with heat shock, which we confirmed by western blot. Therefore, while we can make no direct claims as to the functional impact of ubiquitination itself, we can firmly conclude that protein misfolding (for which ubiquitination serves as a readout) decreases tension-generating capacity at the sarcomere.

4. In Figure 6, why addition of BAG3 rather decreases ubiquitination of the myofilament despite the fact that it promotes CASA assembly at the myofilament in mice?

Response 2D. We thank the reviewer for this question, which we have done our best to clarify through textual revision. Our data support that BAG3 is required for removal of misfolded proteins from the sarcomere. In the mouse HF model, the CASA complex aggregated in the myofilament fraction indicating that the targeting of the complex to the sarcomere in response to stress was maintained. However, the clearance or turnover of the CASA complex by autophagy is impaired at this stage in heart failure progression, which leads to a build up of ubiquitinated misfolded proteins as indicated by our data. Increasing BAG3 expression, which our data suggests restores/increases the CASA client clearance at the sarcomere, would of course increase the ubiquitination of sarcomere protein initially. However, over time, this increased CASA activity would lead to these ubiquitinated proteins being degraded by autophagy. We expressed AAV9/BAG3 for 4 weeks in the HF mice, and thus at the time-point we observe, the ubiquitination at the myofilament is down toward Sham levels due to restored clearance of misfolded proteins by the CASA complex.

5. Why is the assembly of the CASA complex in the myofilament fraction opposite (one disappears and the other stays) between human DCM and mouse heart failure?

Response 2E. We appreciate the reviewer's distinction that the myofilament fraction BAG3 expression in the human DCM samples is decreased, while CASA protein expression was increased in the mouse HF model. In our revised manuscript, we have added more discussion to interpret these findings and offer an explanation of the discrepancy. Our conclusion based on the data is that the expression of CASA at the myofilament differs depending on the progression of heart failure. In the earlier stages (mouse), the proteotoxic stress-dependent regulation of CASA localization to the sarcomere (which we show in the NRVM study with MG132) is maintained. However, while targeting is maintained, it appears the CASA clearance stalls in the heart failure mice or is occurring at an inadequate rate for the level of protein misfolding/stress. In the human samples, which come from patients who had been living with heart failure for many years, BAG3 expression decreases. Our interpretation is that in the end-stage of heart failure the stress-responsiveness of BAG3/CASA is dysregulated. That protein degradation pathways are dysregulated in end-stage heart failure is supported by numerous studies (Martin and Kirk, *J Mol Cell Cardiol*, 2020).

Whether CASA is targeted to the sarcomere in response to stress but fails to clear misfolded substrates, or the CASA stress response is dysregulated as occurs in the end-stage of heart failure, the net result is that misfolded sarcomere proteins are not be adequately cleared. The ultimate result

is that misfolded proteins remain integrated in the contractile apparatus, as our data supports, and cause significant dysfunction.

6. According to Figure 7A, the effect of AAV-BAG3 upon the level of total BAG3 is negligible. Why can it induce a significant effect?

Response 2F. Thank you. Again, it comes down to clearance. Previous studies have shown that BAG3 is degraded along with the CASA complex and its substrates (Kathage *et al.*, *Biochem et Biophysica Acta*, 2017). So, with an increased turnover of the clients, we would also expect an increased turnover of BAG3 and thus no significant increase or a mild increase in BAG3 expression with the AAV9/BAG3 treatment. We have added to discussion to address this point (which is related to point 4 and 5 as well).

7. In Figure 5J, one cannot say anything about the level of CHIP with $p=0.21$.

Response 2G. We agree with the reviewer and overstated the relationship between decreased BAG3 expression and CHIP in the human samples. This has been addressed in the revised manuscript.

Reviewer #3:

The manuscript has substantially improved and all the concerns have been addressed either experimentally or through textual revision.

Reviewer #4:

Remarkable effort has been made to improve this manuscript...However, there are still several minor issues that you may consider to rephrase during the proof-reading.

1. P3, Line 81-82: "An elevation of dysfunctional proteins incorporated into the sarcomere in heart failure...". These dysfunctional proteins were denatured by mechanical stress; they are existing components of sarcomere. Could "An arrested removal of dysfunctional proteins led to their aggregations in the sarcomere in heart failure" better describe the mechanism of decreased F_{max} ?

Response 4A. We appreciate the reviewer's comment on the phrasing and have incorporated the reviewer's suggestion in the revised manuscript.

2. P5, Line 118: "where lowest BAG3 expression predicted weakest force generation". Both BAG3

expression level and force generation capacity were measured for human DCM samples. Should these two parameters be described as “correlated” instead of “predicted”?

Response 4B. Thank you. We agree that “correlated” better describes the relationship between force-generating capacity and BAG3 expression. We have addressed this in the revised manuscript.

3. The manuscript could benefit from another round of editing to elevate its readability. We noticed a few cases, such as adding “that” after “indicates” on P5 Line 172, moving “by western blot” after “were found” on P5 Line 173, and adding “that” after “suggests” on P5 Line 175.

Response 4C. We appreciate the reviewer’s comment and have incorporated these suggestions and author wording changes throughout to elevate its readability.

4. We understand the author’s argument on not calculating correlation between F_{max} and BAG3 level for the reason that the relationship may not be linear and a modest loss of BAG3 may not be pathologically important. However, we believe that a scatter plot can address the problems and better demonstrate the trend.

Response 4D. Certainly much can be gleaned from multiple ways of organizing the data. At the reviewer’s suggestion we have added scatter plot analysis of these data to the supplemental file, where the relationship between increased BAG3 expression and increased F_{max} is further borne out (**Supplemental Figure S4**).

REVIEWER COMMENTS<

Reviewer #5 (Remarks to the Author):

This is a revised manuscript from Kirk's group (Loyola University Chicago Stritch School of Medicine). They investigate the role of Bcl2-associated athanogene 3 (BAG3) on cardiomyocyte contractile impairment in heart failure. They found that BAG3 plays a critical role for the functional maintenance of the cardiac sarcomere through mediating sarcomere protein turnover.

The authors have been very responsive and have adequately revised the manuscript.

The previous major concerns were wholly addressed:

- 1) They assessed the total protein level for each ubiquitinated protein and found they did not change between non-failing and DCM in the humans or between the sham, HF, and HF/BAG3 models in the mice (new Figures S2 & S6).
- 2) They performed mass spectrometry analysis of the HSP70 myofilament interactome and identified two additional ubiquitin ligases, UBA52 and MIB2 (Supplementary Data 2), and identified seven other E3 ligases in the BAG3 myofilament interactome (Supplementary Data 1). The data show that this process is more complicated than just CHIP mediating all ubiquitination of sarcomere proteins. The new data depict that BAG3 is an essential protein for sarcomere protein turnover.
- 3) They performed a new functional experiment. They took skinned left ventricular cardiac myocytes from the non-failing human samples and incubated them with mild heat shock at 43°C to cause protein denaturation/misfolding. Then they performed force-calcium functional experiments on these myocytes and controls from the same samples (Figure S1). They conclude that protein misfolding decreases tension-generating capacity at the sarcomere.
- 4) They expressed AAV9/BAG3 for four weeks in the heart failure mice. They found that the myofilament's ubiquitination is down toward Sham levels due to restored clearance of misfolded proteins by the CASA complex.
- 5) They clarify that the myofilament fraction BAG3 expression in the human DCM samples is decreased, while CASA protein expression was increased in the mouse heart failure model.
- 6) They corrected that the effect of AAV-BAG3 upon the level of total BAG3 is negligible (Fig 7A)
- 7) They removed the overstated conclusion that the relationship between decreased BAG3 expression and CHIP in the human samples.

I do not have further comments.

REVIEWERS' COMMENTS

Reviewer #4 (Remarks to the Author):

After the latest round of revision, the quality of this manuscript has been considerably improved. The authors' conclusions on their findings are supported by their data. The current focus on BAG3-mediated sarcomere protein turnover is appropriate and data collected in this investigation is of excellent quality. The new MS/MS analysis of the HSPB8 myofilament interactome is an elevation of their discovery.